# Rethinking Spectral Augmentation for Contrast-based Graph Self-Supervised Learning

**Xiangru Jian**[*]                                                                                    *xiangru.jian@uwaterloo.ca*
*Cheriton School of Computer Science*
*University of Waterloo*

**Xinjian Zhao**[*]                                                                                    *xinjianzhao1@link.cuhk.edu.cn*
*School of Data Science*
*The Chinese University of Hong Kong, Shenzhen*

**Wei Pang**[*]                                                                                    *w3pang@uwaterloo.ca*
*Cheriton School of Computer Science*
*University of Waterloo*
*Vector Institute*

**Chaolong Ying**                                                                                    *chaolongying@link.cuhk.edu.cn*
*School of Data Science*
*The Chinese University of Hong Kong, Shenzhen*

**Yimu Wang**                                                                                    *yimu.wang@uwaterloo.ca*
*Cheriton School of Computer Science*
*University of Waterloo*

**Yaoyao Xu**                                                                                    *xuyaoyao@cuhk.edu.cn*
*School of Data Science*
*The Chinese University of Hong Kong, Shenzhen*

**Tianshu Yu**[†]                                                                                    *yutianshu@cuhk.edu.cn*
*School of Data Science*
*The Chinese University of Hong Kong, Shenzhen*

**Reviewed on OpenReview:** *https://openreview.net/forum?id=HjpD5kpfa3*

## Abstract

The recent surge in contrast-based graph self-supervised learning has prominently featured an intensified exploration of spectral cues. Spectral augmentation, which involves modifying a graph's spectral properties such as eigenvalues or eigenvectors, is widely believed to enhance model performance. However, an intriguing paradox emerges, as methods grounded in seemingly conflicting assumptions regarding the spectral domain demonstrate notable enhancements in learning performance. Through extensive empirical studies, we find that simple edge perturbations - random edge dropping for node-level and random edge adding for graph-level self-supervised learning - consistently yield comparable or superior performance while being significantly more computationally efficient. This suggests that the computational overhead of sophisticated spectral augmentations may not justify their practical benefits. Our theoretical analysis of the InfoNCE loss bounds for shallow GNNs further supports this observation. The proposed insights represent a significant leap forward in the field, potentially refining the understanding and implementation of graph self-supervised learning.

---

[*]Xiangru Jian, Xinjian Zhao, and Wei Pang contributed equally to this paper.
[†]Corresponding author

# 1 Introduction

In recent years, graph learning has emerged as a powerhouse for handling complex data relationships in multiple fields, offering vast potential and value, particularly in domains such as data mining (Hamilton et al., 2017), computer vision (Xu et al., 2017), network analysis (Chen et al., 2020b), and bioinformatics (Jin et al., 2018). However, limited labels make graph learning challenging to apply in real-world scenarios. Inspired by the great success of Self-Supervised Learning (SSL) in other domains (Devlin et al., 2018; Chen et al., 2020a), Graph Self-Supervised Learning (Graph SSL) has made rapid progress and has shown promise by achieving state-of-the-art performance on many tasks (Xie et al., 2022), where **C**ontrast-based **G**raph **SSL** (**CG-SSL**) are most dominant (Liu et al., 2023). This type of method is grounded in the concept of mutual information (MI) maximization. The primary goal is to maximize the estimated MI between augmented instances of the same object, such as nodes, subgraphs, or entire graphs. Among the new developments in **CG-SSL**, approaches inspired by graph spectral methods have garnered significant attention. A prevalent conviction is that spectral information, including the eigenvalues and eigenvectors of the graph's Laplacian, plays a crucial role in enhancing the efficacy of **CG-SSL** (Liu et al., 2022a; Ko et al., 2023; Lin et al., 2023; Yang et al., 2023; Chen et al., 2024).

In general, methods in **CG-SSL** can be categorized into two types based on whether augmentation is performed on the input graph to generate different views (Chen et al., 2024). i.e. augmentation-based and augmentation-free methods. Of the two, the augmentation-based methods are more predominant and widely studied (Hassani & Khasahmadi, 2020; Liu et al., 2023; You et al., 2020; Liu et al., 2022a; Lin et al., 2023; Yang et al., 2023). Specifically, spectral augmentation has received significant attention, as it modifies a graph's spectral properties. This approach is believed to enhance model performance, aligning with the proposed importance of spectral information in **CG-SSL**. However, there seems no consensus on the true effectiveness of spectral information in the previous works proposing and studying spectral augmentation. SpCo (Liu et al., 2022a) introduces the general graph augmentation (GAME) rule, which suggests that the difference in high-frequency parts between augmented graphs should be larger than that of low-frequency parts. SPAN (Lin et al., 2023) contends that effective topology augmentation should prioritize perturbing sensitive edges that have a substantial impact on the graph spectrum. Therefore, a principled augmentation method is designed by directly maximizing spectral change with a certain perturbation budget, without mentioning any specific domain of spectrum. GASSER (Yang et al., 2023) selectively perturbs graph structures based on spectral cues to better maintain the required invariance for contrastive learning frameworks. Specifically, it aims to augment the graphs to preserve task-relevant frequency components and perturb the task-irrelevant ones with care. While all three related methods are augmentation-based and share in the set of **CG-SSL** frameworks like GRACE (Zhu et al., 2020) and MVGRL (Hassani & Khasahmadi, 2020), a contradiction emerges among these related works on spectral augmentation: while SPAN advocates for **maximizing the distance** between the spectrum of augmented graphs regardless of spectral domains, SpCo and GASSER argue for the **preservation** of specific spectral components and domains during augmentation. The consistent performance gain derived from opposing methical designs naturally raises our concern:

- *Are spectral augmentations necessary in contrast-based graph SSL?*

Given the question, this study aims to critically evaluate the effectiveness and significance of spectral augmentation in contrast-based graph SSL frameworks (**CG-SSL**). With evidence-supported claims and findings in the following sections, we show that despite their computational complexity, sophisticated spectral augmentations do not demonstrate clear advantages over simple edge perturbations. Our extensive experiments reveal that straightforward edge perturbations consistently achieve superior performance while being significantly more computationally efficient. Our theoretical analysis on the InfoNCE loss bounds for shallow GNNs provides additional insights into understanding this phenomenon and supports our claims. We elaborate on our findings through a series of studies carried out in the following efforts:

1. In Sec. 4, we demonstrate that shallow networks consistently achieve better performance in **CG-SSL**, analyze their inherent limitations in capturing global spectral information, and provide theoretical bounds on the InfoNCE loss that help explain the limited benefits of sophisticated spectral augmentations compared to simple edge perturbation.

2. In Sec 5, we claim that simple edge perturbation techniques, like adding edges to or dropping edges from the graph, not only compete well but often outperform spectral augmentations, without any significant help from spectral cues. To support this,

   **(a)** In Sec. 6, overall model performance on test accuracy with four state-of-the-art frameworks on both node- and graph-level classification tasks support the superiority of simple edge perturbation.
   **(b)** Studies in Sec. 7.1 reveal the indistinguishability between the average spectrum of augmented graphs from edge perturbation with optimal parameters on different datasets, no matter how different that of original graphs is, indicating GNN encoders can hardly learn spectral information from augmented graphs. That is to say, edge perturbations can not benefit from spectral information.
   **(c)** In Sec. 7.2, we analyze the effectiveness of state-of-the-art spectral augmentation baseline (*i.e.*, SPAN) by perturbing edges to alter the spectral characteristics of augmented graphs from simple edge perturbation augmentation and examining the impact on model performance. As it turns out, the results show no performance degradation, indicating the spectral information contained in the augmentation is not significant to the model performance.
   **(d)** In Appendix E.3, statistical analysis is carried out to argue that the major reason edge perturbation works well is not because of the spectral information as they are statistically not the key factor on model performance.

## 2 Related work

**Contrast-based Graph Self-Supervised (CG-SSL). CG-SSL** learning alleviates the limitations of supervised learning, which heavily depends on labeled data and often suffers from limited generalization (Liu et al., 2022b). This makes it a promising approach for real-world applications where labeled data is scarce. **CG-SSL** applies a variety of augmentations to the training graph to obtain augmented views. These augmented views, which are derived from the same original graph, are treated as positive sample pairs or sets. The key objective of **CG-SSL** is to maximize the mutual information between these views to learn robust and invariant representations. However, directly computing the mutual information of graph representations is challenging. Hence, in practice, **CG-SSL** frameworks aim to maximize the lower bound of mutual information using different estimators such as InfoNCE (Gutmann & Hyvärinen, 2010), Jensen-Shannon (Nowozin et al., 2016), and Donsker-Varadhan (Belghazi et al., 2018). For instance, frameworks like GRACE (Zhu et al., 2020), GCC (Qiu et al., 2020), and GCA (Zhu et al., 2021b) utilize the InfoNCE estimator as their objective function. On the other hand, MVGRL (Hassani & Khasahmadi, 2020) and InfoGraph (Sun et al., 2019) adopt the Jensen-Shannon estimator. Some **CG-SSL** methods explore alternative principles. G-BT (Bielak et al., 2022) extends the redundancy-reduction principle by decorrelating representations between two augmented views to prevent feature collapse. BGRL (Thakoor et al., 2021) adopts a momentum-driven Siamese architecture, using node feature masking and edge modification as augmentations to maximize mutual information between online and target network representations.

**Graph Augmentations in CG-SSL.** Beyond the choice of objective functions, another crucial aspect of augmentation-based methods in **CG-SSL** is the selection of augmentation techniques. Early work by (Zhu et al., 2020) and (You et al., 2020) introduced several domain-agnostic heuristic graph augmentation for **CG-SSL**, such as edge perturbation, attribute masking, and subgraph sampling. These straightforward and effective methods have been widely adopted in subsequent **CG-SSL** frameworks due to their demonstrated success (Thakoor et al., 2021; Yu et al., 2024). However, these domain-agnostic graph augmentations often lack interpretability, making it difficult to understand the exact impact of these augmentations on the graph structure and learning outcomes. To address this issue, MVGRL (Hassani & Khasahmadi, 2020) introduces graph diffusion as an augmentation strategy, where the original graph provides local structural information and the diffused graph offers global context. Moreover, three spectral augmentation methods–SpCo (Liu et al., 2022a), GASSER (Yang et al., 2023), and SPAN (Lin et al., 2023)–stand out by offering design principles based on spectral graph theory, focusing on how to enhance **CG-SSL** performance through spectral manipulations. However, our explorations show that these methods are unable to consistently outperform heuristic graph augmentations such as edge perturbation (DROPEDGE or ADDEDGE) in terms of performance under fair comparisons, and thus the design principles of graph augmentation still require further validation.

## 3 Preliminary study

**Contrast-based graph self-supervised learning framework. CG-SSL** captures invariant features of a graph by generating multiple views (typically two) through augmentations and then maximizing the mutual information between these views (Xie et al., 2022). This approach is ultimately used to improve performance on various downstream tasks. Following previous work (Wu et al., 2021; Liu et al., 2022b; Xie et al., 2022), we first denote the generic form of the augmentation $\mathcal{T}$ and objective functions $\mathcal{L}_{cl}$ of graph contrastive learning. Given a graph $\mathcal{G} = (\mathbf{A}, \mathbf{X})$ with adjacency matrix $\mathbf{A}$ and feature matrix $\mathbf{X}$, the augmentation is defined as the transformation function $\mathcal{T}$. In this paper, we are mainly concerned with topological augmentation, in which feature matrix $\mathbf{X}$ remains intact:

$$\widetilde{\mathbf{A}}, \widetilde{\mathbf{X}} = \mathcal{T}(\mathbf{A}, \mathbf{X}) = \mathcal{T}(\mathbf{A}), \mathbf{X} \tag{1}$$

In practice, two augmented views of the graph are generated, denoted as $\mathcal{G}^{(1)} = \mathcal{G}(\mathcal{T}_1(\mathbf{A}, \mathbf{X}))$ and $\mathcal{G}^{(2)} = \mathcal{G}(\mathcal{T}_2(\mathbf{A}, \mathbf{X}))$. The objective of GCL is to learn representations by minimizing the contrastive loss $\mathcal{L}_{cl}$ between the augmented views:

$$\theta^*, \phi^* = \underset{\theta, \phi}{\arg\min} \mathcal{L}_{cl} \left( p_\phi \left( f_\theta \left( \mathcal{G}^{(1)} \right), f_\theta \left( \mathcal{G}^{(2)} \right) \right) \right), \tag{2}$$

where $f_\theta$ represents the graph encoder parameterized by $\theta$, and $p_\phi$ is a projection head parameterized by $\phi$. The goal is to find the optimal parameters $\theta^*$ and $\phi^*$ that minimize the contrastive loss.

In this paper, we utilize four prominent **CG-SSL** frameworks to study the effect of spectral: MVGRL, GRACE, BGRL, and G-BT. MVGRL introduces graph diffusion as augmentation, while the other three frameworks use edge perturbation as augmentation. Each framework employs different strategies for its contrastive loss functions. MVGRL and GRACE use the Jensen-Shannon and InfoNCE estimators as object functions, respectively. In contrast, BGRL and G-BT adopt the BYOL loss (Grill et al., 2020) and Barlow Twins loss (Zbontar et al., 2021), which are designed to maximize the agreement between the augmented views without relying on negative samples. More details of the loss function can be found in the Appendix C.

**Graph spectrum & Definition and application of spectral augmentation.** We follow the standard definition of graph spectrum in this study, details of which can be found in Appendix B. Among various augmentation strategies proposed to enhance the robustness and generalization of graph neural networks, spectral augmentation has been considered a promising avenue (Lin et al., 2023; Liu et al., 2022a; Bo et al., 2023; Yang et al., 2023). Spectral augmentation typically involves implicit modifications to the eigenvalues of the graph Laplacian, aiming at enhancing model performance by encouraging invariance to certain spectral properties. Among them, SPAN achieved state-of-the-art performance in both node classification and graph classification. In short, SPAN elaborates two augmentation functions, $\mathcal{T}_1$ and $\mathcal{T}_2$, where $\mathcal{T}_1$ maximizes the spectral norm in one view, and $\mathcal{T}_2$ minimizes it in the other view. Subsequently, these two augmentations are implemented in the four **CG-SSL** frameworks mentioned above (Strict definition in Appendix B). The paradigm used by SPAN aims to allow the GNN encoder to focus on robust spectral components and ignore the sensitive edges that can change the spectral drastically when perturbed.

## 4 Limitations of spectral augmentations

**Limitations of shallow GNN encoders in capturing spectral information.** Multiple previous studies indicate that shallow, rather than deep, GNN encoders can be effective in graph self-supervised learning. This might be the result of overfitting commonly witnessed in standard GNN tasks. We have also carried out many empirical studies with a range of **CG-SSL** frameworks and augmentations to support this idea in contrast-based graph SSL. As the most commonly applied GNN encoder in CG-SSL (You et al., 2020; Yu et al., 2024; Guo et al., 2024; Lin et al., 2024), an empirical study on the relationship between the depth of GCN encoder and learning performance is conducted and results are presented in Fig. 1. From that, we can conclude that shallow GCN encoders with 1 or 2 layers usually have the best performance. Note that this tendency is not clear on graph-level tasks, which can be partially explained by the beneficial oversmoothing phenomenon present in this context Southern et al. (2024). It suggests that while deep encoders may have

theoretically better expressive power than shallower ones, the limited benefits of deeper GNN architectures in the current **CG-SSL** practice imply that more layers may not bring significant improvements and could even hinder the quality of the learned graph representations.

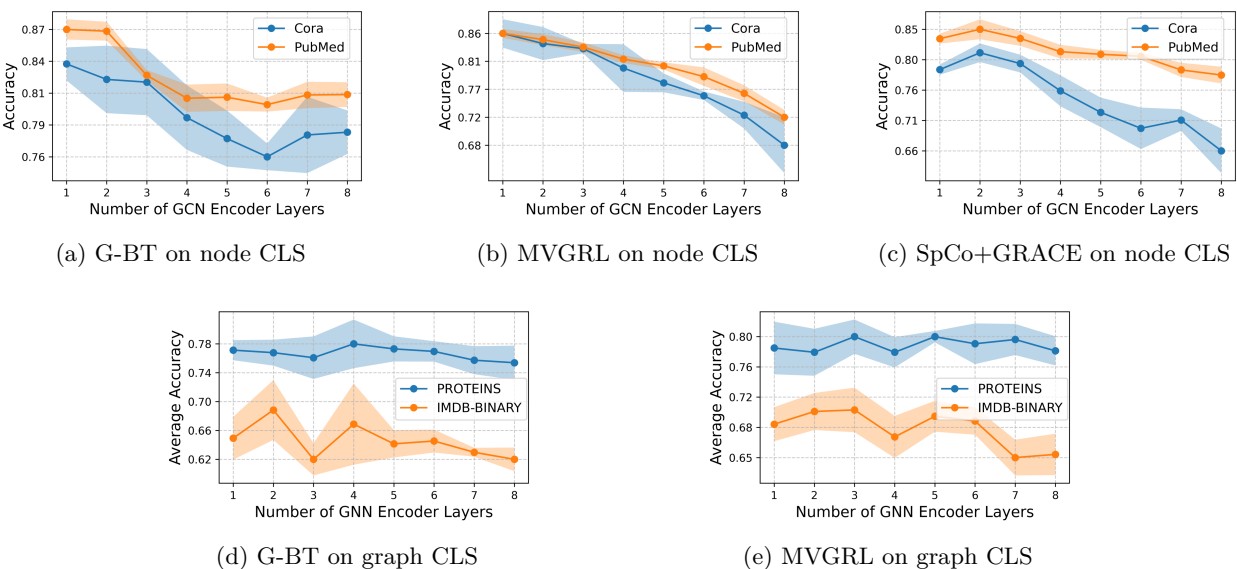

(a) G-BT on node CLS       (b) MVGRL on node CLS       (c) SpCo+GRACE on node CLS

(d) G-BT on graph CLS       (e) MVGRL on graph CLS

Figure 1: Accuracy of **CG-SSL** vs. number of GCN layers on node and graph classification on four datasets. (a) G-BT on node classification. (b) MVGRL on node classification. (c) SpCo+GRACE on node classification. (d) G-BT on graph classification. (e) MVGRL on graph classification. We choose two representative datasets for each task, i.e. CORA and CITESEER for the node-level and PROTEINS and IMDB-BINARY for the graph-level classification. The evaluation protocol, along with dataset details and other experimental settings, are provided in Section 6.1.

By design, most GNN encoders primarily aggregate local neighborhood information through their layered structure, where each layer extends the receptive field by one hop. The depth of a GNN critically determines its ability to integrate information from various parts of the graph. With only a limited number of layers, a GNN's receptive field is restricted to immediate neighborhoods (e.g., 1-hop or 2-hop distances). This limitation severely constrains the network's ability to assimilate and leverage broader graph topologies or global features that are essential for encoding the spectral properties of the graph, given the definition of the graph spectrum.

**Limited implications for spectral augmentation in CG-SSL.** Given the inherent limitations of shallow GNNs in capturing spectral information, the utility of spectral augmentation techniques in graph self-supervised learning settings warrants scrutiny. While spectral augmentation techniques modify the graph's spectral components (e.g., eigenvalues and eigenvectors) to enrich the training process, their benefits may be limited if the primary encoder—a shallow GNN—cannot effectively process these spectral properties. To formally validate this intuition, we establish the following theoretical analysis.

**Theoretical Analysis of InfoNCE Bounds.** To better understand the effectiveness of augmentations in shallow GNNs, we derive the following theoretical bounds on InfoNCE loss:

**Theorem 1** (InfoNCE Loss Bounds)**.** *Given a graph $\mathcal{G}$ with minimum degree $d_{\min}$ and maximum degree $d_{\max}$, and its augmentation $\mathcal{G}'$ with local topological perturbation strength $\delta$, for a $k$-layer GNN with ReLU activation and weight matrices satisfying $\left\| \mathbf{W}^{(l)} \right\|_2 \leq L_W$, and assuming that the embeddings are normalized*

$(\|\mathbf{z}_v\| = \|\mathbf{z}'_v\| = 1)$, the InfoNCE loss satisfies with high probability:

$$-\log\left(\frac{e^{1/\tau}}{e^{1/\tau} + (n-1)e^{-\epsilon'/\tau}}\right) \leq \mathcal{L}_{InfoNCE}(\mathcal{G}, \mathcal{G}') \leq -\log\left(\frac{e^{\left(1-\frac{\epsilon^2}{2}\right)/\tau}}{e^{\left(1-\frac{\epsilon^2}{2}\right)/\tau} + (n-1)e^{\epsilon'/\tau}}\right), \qquad (3)$$

where $\epsilon$ is as defined in Lemma 5 and $\epsilon'$ is as defined in Lemma 6. Detailed descriptions of notations can be found in Table 6. The proof of Theorem 1 and all related Lemmas can be found in Appendix D.

This theoretical result reveals that the InfoNCE loss naturally stays within a narrow interval given a perturbation strength, regardless of augmentation complexity. Such a finding helps explain why sophisticated spectral augmentations may not significantly outperform simple ones in shallow architectures. While the potential benefits of coupling spectral augmentations with deeper GNN architectures remain an open question.

**Numerical Estimation and Interpretation.** To illustrate the derived bounds, we provide a numerical estimation of the upper and lower bounds based on realistic parameters for 1-layer GNNs (which is the case for best performance for several benchmarks presented in Sec. 6 later). As detailed in Appendix D.5, the parameters were chosen using typical graph augmentation settings and realistic assumptions about $\epsilon$ and $\epsilon'$ derived from Lemma 5 and Lemma 6. The resulting bounds on the InfoNCE loss are: Lower bound: 4.7989, Upper bound: 5.4497. The difference between the two bounds is $5.4497 - 4.7989 = 0.6508$, indicating that the InfoNCE loss remains tightly constrained under these settings. This small interval suggests that shallow GNNs cannot fully utilize complex spectral augmentations, as their expressive capacity limits the potential variation in mutual information captured from augmented views. Our analysis reveals a critical insight: the limited efficacy of spectral augmentation stems from the inability of shallow GNNs to effectively capture and leverage the spectral properties of a graph. Instead, the learning outcomes are more directly influenced by simpler factors, such as the strength of edge perturbations. These findings reinforce the practicality of straightforward augmentation methods like edge dropping and adding, which perform comparably or better in this constrained theoretical setting.

## 5 Edge perturbation is all you need

So far, our findings indicate that spectral augmentation is not particularly effective in contrast-based graph self-supervised learning. It may suggest that spectral augmentation essentially amounts to random topology perturbation, based on inconsistencies in previous studies (Lin et al., 2023; Liu et al., 2022a; Yang et al., 2023) and the theoretical insight that a shallow encoder can hardly capture spectral properties. In fact, most of the spectral augmentations basically perform edge perturbations on the graph with some targeted directions. Since we preliminarily conclude that it is quite difficult for those augmentations to benefit from the spectral properties of graphs, it is very intuitive to hypothesize that edge perturbation itself matters in the learning process.

Consequently, we are turning back to **E**dge **P**erturbation (**EP**), a more straightforward and proven method for augmenting graph data. The two primary methods of edge perturbation are DROPEDGE and ADDEDGE. We want to claim that edge perturbation has a better performance than spectral augmentations and prove empirically that none of them actually or even can benefit much from any spectral information and properties. Also, we demonstrate edge perturbation is much more efficient in practical applications for both time and space sake, where spectral operations are almost infeasible. Overall, we will support the idea with evidence in the following sections that simple edge perturbation is not only good enough but even very optimal in **CG-SSL** compared to spectral augmentations.

Edge perturbation involves modifying the topology of the graph by either removing or adding edges at random. We detail the two main types of edge perturbation techniques used in our frameworks: edge dropping and edge adding.

**DropEdge.** Edge dropping is the process of randomly removing a subset of edges from the original graph to create an augmented view. Adopting the definition from (Rong et al., 2020), let $\mathcal{G} = (\mathbf{A}, \mathbf{X})$ be the original

graph with adjacency matrix $\mathbf{A}$. We introduce a mask matrix $\mathbf{M}$ of the same dimensions as $\mathbf{A}$, where each entry $M_{ij}$ follows a Bernoulli distribution with parameter $1-p$ (denoted as the drop rate). The edge-dropped graph $G'$ is then obtained by element-wise multiplication of $\mathbf{A}$ with $\mathbf{M}$ (where $\odot$ denotes the Hadamard product):

$$\mathbf{A}' = \mathbf{A} \odot \mathbf{M} \tag{4}$$

**AddEdge.** Edge adding involves randomly adding a subset of new edges to the original graph to create an augmented view. Let $\mathbf{N}$ be an adding matrix of the same dimensions as $\mathbf{A}$, where each entry $N_{ij}$ follows a Bernoulli distribution with parameter $q$ (denoted as the add rate), and $N_{ij} = 0$ for all existing edges in $\mathbf{A}$. The edge-added graph $\mathcal{G}''$ is obtained by adding $\mathbf{N}$ to $\mathbf{A}$:

$$\mathbf{A}'' = \mathbf{A} + \mathbf{N} \tag{5}$$

These two operations ensure that the augmented views $\mathcal{G}^{(1)}$ and $\mathcal{G}^{(2)}$ have modified adjacency matrices $\mathbf{A}'$ and $\mathbf{A}''$ respectively, which are used to generate contrastive views while preserving the feature matrix $\mathbf{X}$.

### 5.1 Advantage of edge perturbation over spectral augmentations

Edge perturbation offers several key advantages over spectral augmentation, making it a more effective and practical choice for **CG-SSL**. Compared to spectrum-related augmentations, it has three major advantages.

**Theoretically intuitive.** Edge perturbation is inherently simpler and more intuitive. It directly modifies the graph's structure by adding or removing edges, which aligns well with the shallow GNN encoders' strength in capturing local neighborhood information. Given that shallow GNNs have a limited receptive field, they are better suited to leveraging the local structural changes introduced by edge perturbation rather than the global changes implied by spectral augmentation.

**Significantly better efficiency.** Edge perturbation methods such as DropEdge and AddEdge are computationally efficient. Unlike spectral augmentation, which requires costly eigenvalue and eigenvector computations, edge perturbation can be implemented with basic graph operations. This efficiency translates to faster training and inference times, making it more suitable for large-scale graph datasets and real-time applications. As shown in Table 1, the time and space complexity of spectrum-related calculations are several orders of magnitude higher than those of simple edge perturbation operations. This makes spectrum-related calculations impractical for large datasets typically encountered in real-world applications.

Table 1: Time and space complexity of different methods (Empirical Time is on PubMed dataset)

| Method | Time Complexity | Space Complexity | Empirical Time (s/epoch) |
|---|---|---|---|
| Spectrum calculation | $O(n^3)$ | $O(n^2)$ | 26.435 |
| DropEdge | $O(m)$ | $O(m)$ | 0.140 |
| AddEdge | $O(m)$ | $O(m)$ | 0.159 |

**Optimal learning performance.** Most importantly and directly, our comprehensive empirical studies indicate that edge perturbation methods lead to significant improvements in model performance, as presented and analyzed in Sec. 6. From the results there, the conclusion can be drawn that the performance of the proposed augmentations is not only better than those of spectral augmentations but also matches or even surpasses the performance of other strong benchmarks.

These advantages position edge perturbation as a robust and efficient method for graph augmentation in self-supervised learning. In the following section, we will present our experimental analysis, demonstrating the accuracy gains achieved through edge perturbation methods.

## 6 Experiments on SSL performance

### 6.1 Experimental Settings

**Task and Datasets.** We conducted extensive experiments for node-level classification on seven datasets: Cora, CiteSeer, PubMed (Kipf & Welling, 2016), Photo, Computers (Shchur et al., 2018), Coauthor-

CS, and COAUTHOR-PHY. These datasets include various types of graphs, such as citation networks, co-purchase networks, and co-authored networks. Note that we do not include huge-scale datasets like OGBN (Hu et al., 2021) for the high complexity of spectral augmentations. While both DROPEDGE and ADDEDGE have linear complexity that can easily run on those huge datasets, no spectral augmentation can scale to them. Additionally, we carried out graph-level classification on five datasets from the TUDataset collection (Morris et al., 2020), which include biochemical molecules and social networks. More details of these datasets be found in Appendix A.

**Baselines.** We conducted experiments under four **CG-SSL** frameworks: MVGRL, GRACE, G-BT, and BGRL (mentioned in Sec 3), using DROPEDGE, ADDEDGE, and SPAN (Lin et al., 2023) as augmentation strategies. Note that there are only three very relevant studies on spectral augmentation strategies of **CG-SSL** to the authors' best knowledge, i.e., SPAN, SpCo (Liu et al., 2022a) and GASSER (Yang et al., 2023). Among them, GASSER does not have open-sourced code so we are not able to reproduce any related results, but we try our best to directly adopt the best performance reported in that study to ensure comparison is possible. Also, SpCo is only applicable to node-level tasks and its implementation is not robust enough to generalize to all the node-level datasets and **CG-SSL** frameworks. Therefore, we manage to include the results of all the settings that it is feasible to do, which is its original setting and the combination of GRACE and it. Given the infeasibility and inaccessibility of the two, we selected SPAN as a major baseline because it is robust and general enough to all the datasets and experimental settings while allowing the modular plug-and-play integration of edge perturbation methods, enabling a very direct angle to evaluate the effectiveness of the spectral augmentations compared to much simpler alternatives. Besides the major baselines mentioned above, other related ones are added to clearly and comprehensively benchmark our work. For MVGRL, we also compared its original PPR augmentation. For the node classification task, we use GCA (Zhu et al., 2021b), GMI (Peng et al., 2020), DGI (Velickovic et al., 2019), CCA-SSG (Zhang et al., 2021) and SpCo (Liu et al., 2022a) as baselines. For the graph classification task, we use RGCL (Li et al., 2022) and GraphCL (You et al., 2020) as baselines. Detailed experimental configurations are in Appendix A.

**Evaluation Protocol.** We adopt the evaluation and split scheme from previous works (Veličković et al., 2019; Zhang et al., 2023; Lin et al., 2023). Each GNN encoder is trained with self-supervised learning. After training, we freeze the encoder and extract embeddings for all nodes or graphs. Finally, we train a simple linear classifier using the labels from the training/validation set and test it with the testing set. The accuracy of classification on the testing set shows how good the learned representations are. For the node classification task nodes are randomly divided into 10%/10%/80% for training, validation, and testing, and for graph classification datasets, graphs are randomly divided into 80%/10%/10% for training, validation, and testing.

## 6.2 Experimental results

We present the prediction accuracy of the node classification and graph classification tasks in Table 2 and Table 3, respectively. Our comprehensive analysis reveals distinct patterns in the effectiveness of different augmentation strategies across these two task types. For node classification, DROPEDGE consistently achieves the best performance across multiple datasets and **CG-SSL** frameworks, demonstrating superior robustness and consistency. While ADDEDGE also achieves competitive accuracy, DROPEDGE stands out in this area. In graph classification, ADDEDGE frequently achieves the best performance across multiple datasets and **CG-SSL** frameworks, showing superior and more consistent results. The effectiveness of ADDEDGE in graph classification may be attributed to 'beneficial oversmoothing' as proposed by Southern et al. (2024). In graph-level tasks, the convergence of node features to a common representation aligned with the global output can be advantageous. By potentially increasing graph density and the proportion of positively curved edges (Nguyen et al., 2023), ADDEDGE might facilitate this beneficial effect in graph classification tasks. Notably, all the results from SPAN as well as GASSER and SpCo generally underperform relative to both DROPEDGE and ADDEDGE while also encountering scalability issues on larger datasets and suffering from a high overhead of training time.

Table 2: Node classification. Results of baselines with '†' are adopted directly from previous works. MV-GRL+PPR is the original setting of MVGRL. The best results in each cell are highlighted in grey. The best results overall are highlighted with **bold and underline**. Metric is accuracy (%).

| Model | Cora | CiteSeer | PubMed | Photo | Computers | Coauthor-CS | Coauthor-Phy |
|---|---|---|---|---|---|---|---|
| GCA† | 83.67 ± 0.44 | 71.48 ± 0.26 | 78.87 ± 0.49 | 92.53 ± 0.16 | 88.94 ± 0.15 | 93.10 ± 0.01 | 95.68 ± 0.05 |
| GMI† | 83.02 ± 0.33 | 72.45 ± 0.12 | 79.94 ± 0.25 | 90.68 ± 0.17 | 82.21 ± 0.31 | 91.08 ± 0.56 | — |
| DGI† | 82.34 ± 0.64 | 71.85 ± 0.74 | 76.82 ± 0.61 | 91.61 ± 0.22 | 83.95 ± 0.47 | 92.15 ± 0.63 | 94.51 ± 0.52 |
| CCA-SSG† | 84.20 ± 0.40 | 73.10 ± 0.30 | 81.60 ± 0.40 | 93.14 ± 0.14 | 88.74 ± 0.28 | 93.31 ± 0.22 | 95.38 ± 0.06 |
| SpCo | 83.78 ± 0.70 | 71.82 ± 1.26 | 80.86 ± 0.43 | — | — | — | — |
| GASSER† | 85.27 ± 0.10 | 75.41 ± 0.84 | 83.00 ± 0.61 | 93.17 ± 0.31 | 88.67 ± 0.15 | — | — |
| MVGRL + PPR | 83.53 ± 1.19 | 71.56 ± 1.89 | 84.13 ± 0.26 | 88.47 ± 1.02 | 89.84 ± 0.12 | 90.57 ± 0.61 | OOM |
| MVGRL + DropEdge | 84.31 ± 1.95 | 74.85 ± 0.73 | 85.62 ± 0.45 | 89.28 ± 0.95 | **90.43 ± 0.33** | 93.20 ± 0.81 | 95.70 ± 0.28 |
| MVGRL + AddEdge | 83.21 ± 1.65 | 73.65 ± 1.60 | 84.86 ± 1.19 | 87.15 ± 1.36 | 87.59 ± 0.53 | 92.91 ± 0.65 | 95.33 ± 0.23 |
| MVGRL +SPAN | 84.57 ± 0.22 | 73.65 ± 1.29 | 85.21 ± 0.81 | 92.33 ± 0.99 | 88.75 ± 0.20 | 92.25 ± 0.76 | OOM |
| MVGRL + GASSER† | 80.36 ± 0.05 | 74.48 ± 0.73 | 80.80 ± 0.19 | — | — | — | — |
| G-BT + DropEdge | **86.51 ± 2.04** | 72.95 ± 2.46 | 87.10 ± 1.21 | 93.55 ± 0.60 | 88.66 ± 0.46 | **93.31 ± 0.05** | **96.06 ± 0.24** |
| G-BT + AddEdge | 82.10 ± 1.48 | 66.36 ± 4.25 | 85.98 ± 0.81 | 93.68 ± 0.79 | 87.81 ± 0.79 | 91.98 ± 0.66 | 95.51 ± 0.02 |
| G-BT + SPAN | 84.06 ± 2.85 | 67.46 ± 3.18 | 85.97 ± 0.41 | 91.85 ± 0.22 | 88.73 ± 0.62 | 92.63 ± 0.07 | OOM |
| GRACE + DropEdge | 84.19 ± 2.07 | **75.44 ± 0.32** | **87.84 ± 0.37** | 92.62 ± 0.73 | 86.67 ± 0.61 | 93.15 ± 0.23 | OOM |
| GRACE + AddEdge | 85.78 ± 0.62 | 71.65 ± 1.63 | 85.25 ± 0.47 | 89.93 ± 0.74 | 76.74 ± 0.57 | 92.46 ± 0.25 | OOM |
| GRACE + SPAN | 82.84 ± 0.91 | 67.76 ± 0.21 | 85.11 ± 0.71 | **93.72 ± 0.21** | 88.71 ± 0.06 | 91.72 ± 1.75 | OOM |
| GRACE + GASSER† | 84.10 ± 0.26 | 74.47 ± 0.64 | 83.97 ± 0.52 | — | — | — | — |
| GRACE + SpCo | 81.61 ± 0.75 | 70.83 ± 1.47 | 84.97 ± 1.13 | — | — | — | — |
| BGRL + DropEdge | 83.21 ± 3.29 | 71.46 ± 0.56 | 86.28 ± 0.13 | 92.90 ± 0.69 | 88.68 ± 0.65 | 91.58 ± 0.18 | 95.29 ± 0.19 |
| BGRL + AddEdge | 81.49 ± 1.21 | 69.66 ± 1.34 | 84.54 ± 0.22 | 91.85 ± 0.75 | 86.75 ± 1.15 | 91.78 ± 0.77 | 95.29 ± 0.09 |
| BGRL + SPAN | 83.33 ± 0.45 | 66.26 ± 0.92 | 85.97 ± 0.41 | 91.72 ± 1.75 | 88.61 ± 0.59 | 92.29 ± 0.59 | OOM |

Table 3: Graph classification. Results of baselines with '†' are adopted directly from previous works. MVGRL+PPR is the original setting of MVGRL. The best results in each cell are highlighted in grey. The best results overall are highlighted with **bold and underline**. Metric is accuracy (%).

| Model | MUTAG | PROTEINS | NCI1 | IMDB-BINARY | IMDB-MULTI |
|---|---|---|---|---|---|
| GraphCL† | 86.80 ± 1.34 | 74.39 ± 0.45 | 77.87 ± 0.41 | 71.14 ± 0.44 | 48.58 ± 0.67 |
| RGCL† | 87.66 ± 1.01 | 75.03 ± 0.43 | 78.14 ± 1.08 | 71.85 ± 0.84 | 49.31 ± 0.42 |
| GMCL † | 94.09 ± 6.28 | 77.44 ± 3.83 | 82.99 ± 2.06 | 75.60 ± 3.17 | — |
| MVGRL + PPR | 90.00 ± 5.40 | 78.92 ± 1.83 | **78.78 ± 1.52** | 71.40 ± 4.17 | **52.13 ± 1.42** |
| MVGRL+ SPAN | 93.33 ± 2.22 | 79.81 ± 2.45 | 77.56 ± 1.77 | 75.00 ± 1.09 | 51.20 ± 1.62 |
| MVGRL+ DropEdge | 93.33 ± 2.22 | 78.92 ± 1.33 | 77.81 ± 1.50 | **76.40 ± 0.48** | 51.46 ± 3.02 |
| MVGRL+ AddEdge | **94.44 ± 3.51** | 81.25 ± 3.43 | 77.27 ± 0.71 | 74.00 ± 2.82 | 51.73 ± 2.43 |
| G-BT + SPAN | 90.00 ± 6.47 | **80.89 ± 3.22** | 78.29 ± 1.12 | 65.60 ± 1.35 | 45.60 ± 2.13 |
| G-BT + DropEdge | 92.59 ± 2.61 | 77.97 ± 0.42 | 78.18 ± 0.91 | 73.33 ± 1.24 | 49.11 ± 1.25 |
| G-BT + AddEdge | 92.59 ± 2.61 | 80.64 ± 1.68 | 75.91 ± 0.59 | 73.33 ± 1.24 | 48.88 ± 1.13 |
| GRACE + SPAN | 90.00 ± 4.15 | 79.10 ± 2.30 | 78.49 ± 0.79 | 70.80 ± 3.96 | 47.73 ± 1.71 |
| GRACE + DropEdge | 88.88 ± 3.51 | 78.21 ± 1.92 | 76.93 ± 1.14 | 71.00 ± 3.75 | 47.46 ± 3.02 |
| GRACE + AddEdge | 92.22 ± 4.44 | 80.17 ± 2.21 | 79.97 ± 2.35 | 71.67 ± 2.36 | 49.86 ± 4.09 |
| BGRL + SPAN | 90.00 ± 4.15 | 79.28 ± 2.73 | 78.05 ± 1.62 | 72.40 ± 2.57 | 47.46 ± 4.35 |
| BGRL + DropEdge | 88.88 ± 4.96 | 76.60 ± 2.21 | 76.15 ± 0.43 | 71.60 ± 3.31 | 51.47 ± 3.02 |
| BGRL + AddEdge | 91.11 ± 5.66 | 79.46 ± 2.18 | 76.98 ± 1.40 | 72.80 ± 2.48 | 47.77 ± 4.18 |

## 6.3 Ablation Study

To validate our findings, we conducted a series of ablation experiments on two exemplar datasets, Cora and MUTAG, representing node- and graph-level tasks, respectively. These ablation studies are crucial to rule out potential confounding variables, such as model architectures and hyperparameters, ensuring that our conclusions about the performance of **CG-SSL** are robust and comprehensive.

**Number of Layers of GCN Encoder.** To assess the impact of model depth, we conducted both node-level and graph-level experiments using varying numbers of GCN encoder layers. This analysis is to rule out the possibility that model depth, rather than augmentation strategies, influences the claim. As expected, the results, detailed in Appendix E.1, show that deeper encoders generally lead to worse performance. This

suggests that excessive model complexity may introduce noise or overfitting, diminishing the benefits of spectral information. Therefore, our conclusion still holds tightly.

**Type of GNN Encoder.** While we initially selected GCN to align with the common protocols in previous studies for a fair comparison, we also explored other GNN architectures to ensure our findings are not specific to GCN alone. To further validate our results, we conducted additional experiments using GAT (Veličković et al., 2019) for both node- and graph-level tasks, as well as GPS (Rampášek et al., 2024) for the graph-level task. As reported in Appendix E.2, the performance trends observed with GAT and GPS are consistent with those obtained using GCN. This consistency across different encoder types further supports our conclusion that simple edge perturbation strategies are sufficient, and that spectral augmentation does not significantly enhance performance, regardless of the type of GNN encoder applied.

## 7   The insignificance of Spectral Cues

Given the superior empirical performance of edge perturbations mentioned in Sec. 6, one may still argue whether it is a result of some spectral cues or not, as all the analyses mentioned are not direct evidence of the insignificance of the spectral information in the study. To clarify this, we have three questions to answer, **(1)** Can GNN encoders learn spectral information from augmented graphs produced by edge perturbations? **(2)** Are spectra in spectral augmentation necessary? **(3)** Is spectral information statistically a significant factor in the performance of edge perturbation? Given the questions, we conduct a series of experimental studies to answer them respectively in Sec. 7.1, 7.2, and Appendix E.3.

### 7.1   Degeneration of the spectrum after Edge Perturbation (EP)

Here we want to conduct studies to answer the question of whether the GNN encoders applied can learn spectral information from the augmented graph views produced by **EP**. Therefore, we collect the spectrum of all augmented graphs ever produced along the way of the contrastive learning process of the best framework with the optimal parameter we have in this study, i.e., G-BT + **EP** with best drop rate $p$ or add rate $q$, and calculate the average one for each representative dataset in this study for both node- and graph-level tasks. We find that though the average spectrum of those original graphs is strikingly different, that of augmented graphs is quite similar for node- and graph-level tasks, respectively. This indicates a certain degree of degeneration of the spectra as they are no longer easy to separate after **EP**. Therefore, GNN encoders can hardly learn spectral information and properties between different original graphs from those augmented graph views. Note that, though we have defined some context of frameworks, this result is generally only dependent on the augmentation methods. We will elaborate on both the node-level and graph-level results in this section.

**Node-Level Analysis.** Here, we visualize the distributions of the average spectrum of graphs at the node level using histograms. The spectral distribution for each graph is represented by a sorted vector of its eigenvalues. When referring to the average spectrum, we mean the average over the eigenvalue vectors of each augmented graph. We plot the histograms of different spectra, normalized to show the probability density. Note that eigenvalues are constrained within the range [0, 2], as we adopted the commonly used symmetrical normalization. We analyze the spectral distributions of three node classification datasets: CORA, CITESEER, and COMPUTERS. We compare the average spectral properties of both original and augmented graphs. The augmentation method used is DROPEDGE, applied with optimal parameters identified for the G-BT method. The results of the visualization are presented in Fig. 3. By comparing the spectrum distributions of original graphs for the datasets in Fig. 3a, we can easily distinguish the spectra of the three datasets. This contrasts with the highly overlapped average spectra of all the datasets, indicating the degeneration mentioned. To support this claim, we also present the comparison of the spectra of original and augmented graphs on all three datasets in Fig. 3c, 3d, and 3e, respectively, to show the obvious changes after the edge perturbations.

**Graph-Level analysis.** For graph-level analysis, we basically follow the settings mentioned above in node-level one. The only difference from the node-level task is that we have multiple original graphs with various numbers of nodes, leading to the inconsistent dimensions of the vector of the eigenvalues. Therefore, to provide a more detailed comparison of spectral properties at the graph level, we employ Kernel Density

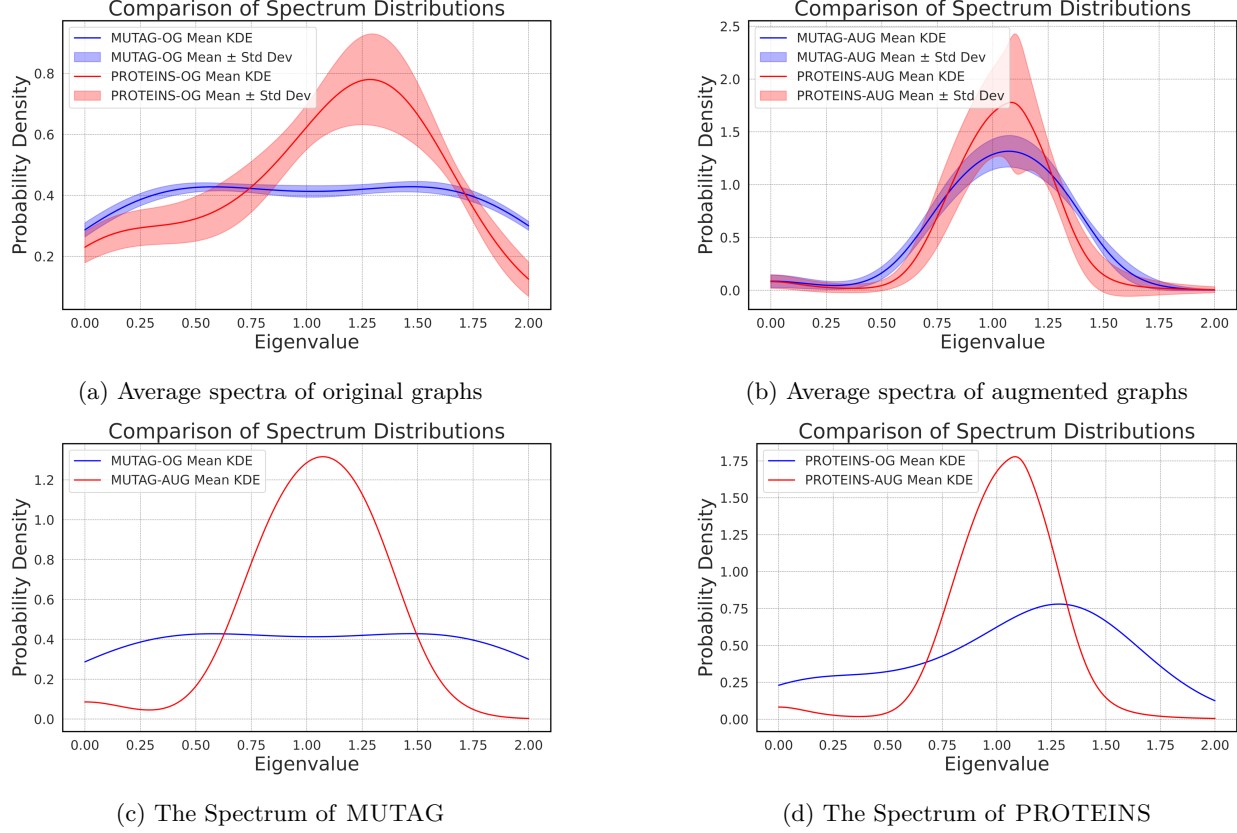

(a) Average spectra of original graphs

(b) Average spectra of augmented graphs

(c) The Spectrum of MUTAG

(d) The Spectrum of PROTEINS

Figure 2: The spectrum distributions of graphs on different graph classification datasets. MUTAG and PROTEINS are chosen as they are well representative of all the node classification datasets. OG means original graph and AUG means augmented graph. The augmentation method is ADDEDGE with the best parameter on G-BT method.

Estimation (KDE) (Parzen, 1962) to interpolate and smooth the distributions of eigenvalues. We compare two groups of graph spectra. Each group's spectra are processed to compute their KDEs, and the mean and standard deviation of these KDEs are calculated.

We analyze the spectral distributions of two node classification datasets: MUTAG and PROTEINS. We compare the average spectral properties of both original and augmented graphs. The augmentation method used is ADDEDGE as it is the better among two **EP** methods, applied with optimal add rate identified for the G-BT method.

Like the results in node-level analysis, in Fig. 2a and 2b, we witness the obvious difference between the average spectra of original graphs while the significant overlap between those of augmented graphs, especially if pay attention to the overlapping of the area created by the standard deviation of KDEs. Again, this contrast is not trivial because of the striking mismatch between the average spectra of original and augmented graphs in both datasets, as presented in Fig. 2c and 2d.

## 7.2 Spectral Perturbation

To further destruct the spectral properties from model performance, we introduce *Spectral Perturbation Augmentor* (SPA) for finer-grained anatomy. SPA performs random edge perturbation with an empirically negligible ratio $r_{SPA}$ to transform the input graph $\mathcal{G}$ into a new graph $\mathcal{G}_{SPA}$, such that $\mathcal{G}$ and $\mathcal{G}_{SPA}$ are close to each other topologically, while being divergent in the spectral space. The spectral divergence $d_{SPA}$ between $\mathcal{G}$ and $\mathcal{G}_{SPA}$ is measured by the $L_2$-distance of the respective spectra. With properly chosen hyperparameters

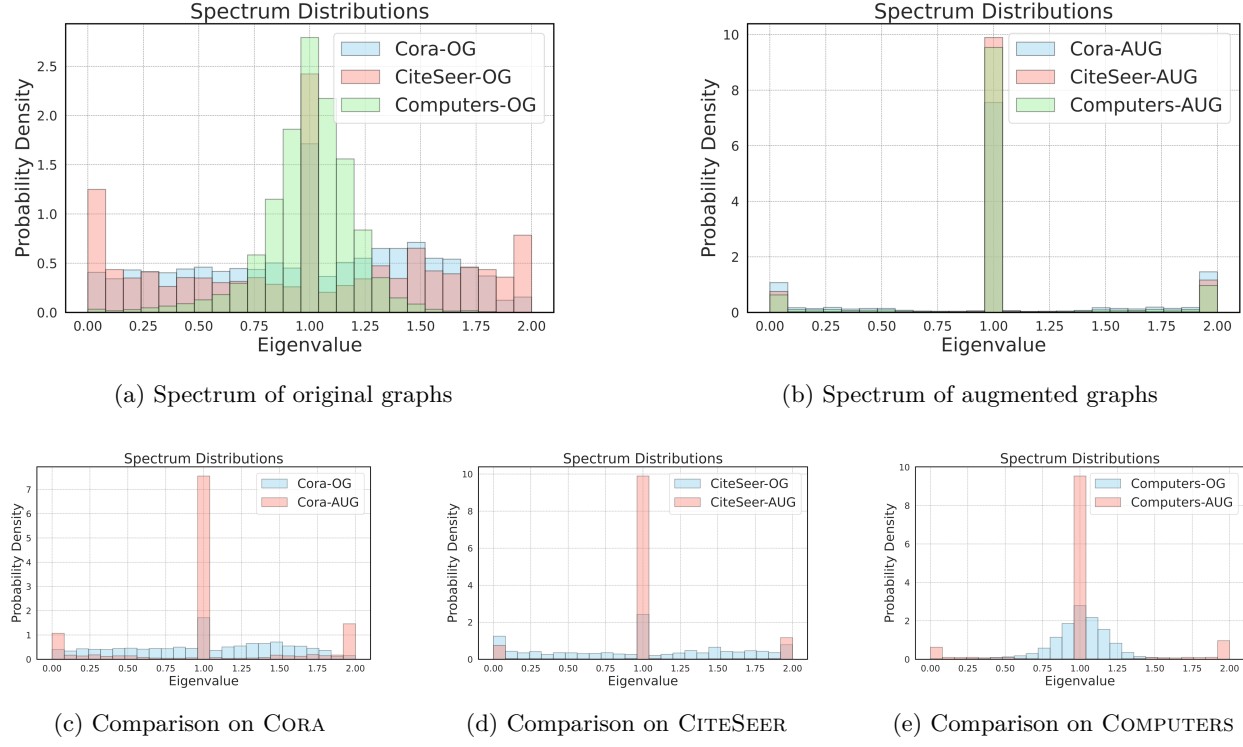

(a) Spectrum of original graphs

(b) Spectrum of augmented graphs

(c) Comparison on CORA

(d) Comparison on CITESEER

(e) Comparison on COMPUTERS

Figure 3: The spectrum distributions of graphs on different node classification datasets. CORA, CITESEER, and COMPUTERS are chosen as they are well representative of all the node classification datasets. OG means original graph and AUG means average augmented graphs. The augmentation method is DROPEDGE with the best parameter on G-BT method.

$r_{SPA}$ and $d_{SPA}$, we view the augmented graph $\mathcal{G}_{SPA}$ as a doppelganger of $\mathcal{G}$ that preserves most of the graph-proximity, with only spectral information eliminated.

**Spectral perturbation on spectral augmentation baselines.** SPAN, being a state-of-the-art spectral augmentation algorithm, demonstrated the correlation between graph spectra and model performance through designated perturbation on spectral priors. However, the effectiveness of simple edge perturbation motivated us to further investigate whether such a relationship is causational.

Specifically, for each pair of SPAN augmented graphs $\mathcal{G}^1, \mathcal{G}^2$, we further augment them into $\mathcal{G}^1_{SPA}, \mathcal{G}^2_{SPA}$ with our proposed SPA augmentor. The SPA-augmented training is performed under the same setup as SPAN, with graphs being SPA-augmented graphs $\mathcal{G}_{SPA}$. Results in Fig 4 show that the effectiveness of graph augmentation can be preserved and, in some cases improved, even if the spectral information is destroyed.

SPAN, along with other spectral augmentation algorithms, can be formulated as an optimization on a parameterized 2-step generative process:

$$s_{SPAN} \sim p_\theta \left( \boldsymbol{S}_{SPAN} \mid \boldsymbol{\mathcal{G}}_0 \right), \qquad \mathcal{G}_{SPAN} \sim p_\phi \left( \boldsymbol{\mathcal{G}}_{SPAN} \mid \boldsymbol{S}_{SPAN} \right) \tag{6}$$

Given the property that $\mathcal{G}_{SPA}$ is topologically close to $\mathcal{G}_{SPAN}$ and the performance function $P = f(\mathcal{G}), \lim_{\mathcal{G} \to \mathcal{G}_{SPAN}} P(\mathcal{G}) = P(\mathcal{G}_{SPAN})$, which indicates the continuity around $\mathcal{G}_{SPAN}$, we make a reasonable assertion that $\mathcal{G}_{SPA}$ comes from the same distribution as $\mathcal{G}_{SPAN}$. However, with their spectral space being enforced to be distant, $\mathcal{G}_{SPA}$ is almost impossible to be sampled from the same spectral augmentation generative process:

$$d_{SPA} \to \infty \implies p_\theta \left( s_{SPA} \mid \boldsymbol{\mathcal{G}}_0 \right) \to 0 \implies p_{\theta,\phi} \left( \mathcal{G}_{SPA} \mid \boldsymbol{\mathcal{G}}_0 \right) \to 0 \tag{7}$$

Although the constrained generative process in Eq. 6 does indicate some extent of causality between spectral distribution $\boldsymbol{S}$ and the spectral-augmented graph distribution $\boldsymbol{\mathcal{G}}_{SPAN}$, our experiment challenges a more

essential and fundamental aspect of such reasoning: such causality exists upon pre-defined generative processes, which does not intrinsically exist in the graph distributions. Even worse, such constrained generative process is incapable of modeling the full distribution of $\boldsymbol{\mathcal{G}}_{SPAN}$ itself. In our experiment setup, all $\mathcal{G}_{SPA}$ serve as strong counter examples.

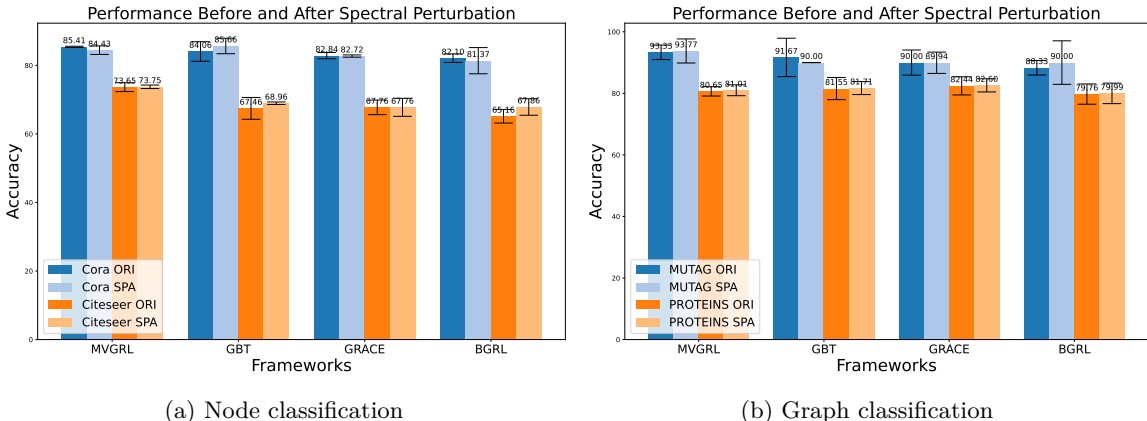

(a) Node classification           (b) Graph classification

Figure 4: Comparison of SPAN performance before and after applying SPA. After severely disrupting the spectral, the performance of SPAN is still comparable to that of the original version.

## 8 Conclusion

In this study, we investigate the effectiveness of spectral augmentation in contrast-based graph self-supervised learning (**CG-SSL**) frameworks to answer the question: *Are spectral augmentations necessary in **CG-SSL***? Our findings indicate that spectral augmentation does not significantly enhance learning efficacy. Instead, simpler edge perturbation techniques, such as random edge dropping for node-level tasks and random edge adding for graph-level tasks, not only compete well but often outperform spectral augmentations. To be specific, we demonstrate that the benefits of spectral augmentation diminish with shallower networks, and edge perturbations yield superior performance in both node- and graph-level classification tasks. Also, GNN encoders struggle to learn spectral information from augmented graphs, and perturbing edges to alter spectral characteristics does not degrade model performance. Furthermore, our theoretical analysis (Theorem 1) reveals that the InfoNCE loss bounds the mutual information achievable by augmentations, highlighting that the relatively limited direct contribution of spectral augmentations compared to simpler edge perturbations, especially when in shallow GNNs. These results challenge the current emphasis on spectral augmentation, advocating for more straightforward and effective edge perturbation techniques in **CG-SSL**, potentially refining the understanding and implementation of graph self-supervised learning.

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

## Contents

# A  Dataset and training configuration

**Datasets.** The node classification datasets used in this paper include the Cora, CiteSeer, and PubMed citation networks (Kipf & Welling, 2016), as well as the Photo and Computers co-purchase networks (Shchur et al., 2018). Additionally, we use the Coauthor-CS and Coauthor-Phy co-author relationship networks. The statistics of node-level datasets are present in Table 4. The graph classification datasets include: The MUTAG dataset, which features seven types of graphs derived from 188 mutagenic compounds; the NCI1 dataset, which contains compounds tested for their ability to inhibit human tumor cell growth; the PROTEINS dataset, where nodes correspond to secondary structure elements connected if they are adjacent in 3D space; and the IMDB-BINARY and IMDB-MULTI movie collaboration datasets, where graphs depict interactions among actors and actresses, with edges denoting their collaborations in films. These movie graphs are labeled according to their genres. The statistics of graph-level datasets are present in Table 5. All datasets can be accessed through PyG library [1]. All experiments are conducted using 8 NVIDIA A100 GPU.

Table 4: Statistics of node classification datasets

| Dataset | #Nodes | #Edges | #Features | #Classes |
|---------|--------|--------|-----------|----------|
| Cora | 2,708 | 5,429 | 1,433 | 7 |
| CiteSeer | 3,327 | 4,732 | 3,703 | 6 |
| PubMed | 19,717 | 44,338 | 500 | 3 |
| Computers | 13,752 | 245,861 | 767 | 10 |
| Photo | 7,650 | 119,081 | 745 | 8 |
| Coauthor-CS | 18,333 | 81,894 | 6,805 | 15 |
| Coauthor-Phy | 34,493 | 247,962 | 8,415 | 5 |

Table 5: Statistics of node classification datasets

| Dataset | #Avg. Nodes | #Avg. Edges | # Graphs | #Classes |
|---------|-------------|-------------|----------|----------|
| MUTAG | 17.93 | 19.71 | 188 | 2 |
| PROTEINS | 39.06 | 72.82 | 1,113 | 2 |
| NCI1 | 29.87 | 32.30 | 4110 | 2 |
| IMDB-BINARY | 19.8 | 96.53 | 1,000 | 2 |
| IMDB-MULTI | 13.0 | 65.94 | 1,500 | 5 |

**Training configuration.** For each **CG-SSL** framework, we implement it based on (Zhu et al., 2021a) [2]. We use the following hyperparameters: the learning rate is set to $5 \times 10^{-4}$, and the node hidden size is set to 512, the number of GCN encoder layer is set $\in \{1, 2\}$. For all node classification datasets, training epochs are set $\in \{50, 100, 150, 200, 400, 1000\}$, and for all graph classification datasets, training epochs are set $\in \{20, 40, ..., 200\}$. To achieve performance closer to the global optimum, we use randomized search to determine the optimal probability of edge perturbation and SPAN perturbation ratio. For Cora and

---

[1] https://pytorch-geometric.readthedocs.io/en/latest/modules/datasets.html
[2] https://github.com/PyGCL/PyGCL

CiteSeer the search is conducted one hundred times, and for all other datasets, it is conducted twenty times. For all graph classification datasets, the batch size is set to 128.

## B Preliminaries of Graph Spectrum and SPAN

Given a graph $\mathcal{G} = (\mathbf{A}, \mathbf{X})$ with adjacency matrix $\mathbf{A}$ and feature matrix $\mathbf{X}$, we introduce some fundamental concepts related to the graph spectrum.

**Laplacian Matrix Spectrum** The Laplacian matrix $\mathbf{L}$ of a graph is defined as:

$$\mathbf{L} = \mathbf{D} - \mathbf{A}$$

where $\mathbf{D}$ is the degree matrix, a diagonal matrix where each diagonal element $D_{ii}$ represents the degree of vertex $i$. The eigenvalues of the Laplacian matrix, known as the Laplacian spectrum, are crucial in understanding the graph's structural properties, such as its connectivity and the number of spanning trees (Chung, 1997).

**Normalized Laplacian Spectrum** The normalized Laplacian matrix $\mathbf{L}_{\mathrm{norm}}$ is given by:

$$\mathbf{L}_{\mathrm{norm}} = \mathbf{D}^{-1/2}\mathbf{L}\mathbf{D}^{-1/2}$$

The eigenvalues of the normalized Laplacian matrix, referred to as the normalized Laplacian spectrum, are often used in spectral clustering (Von Luxburg, 2007) and other applications where normalization is necessary to account for varying vertex degrees.

**SPAN** The core assumption of SPAN is to maximize the consistency of the representations of two views with a large spectrum distance, thereby filtering out edges sensitive to the spectrum, such as edges between clusters. By focusing on more stable structures relative to the spectrum, the objective of SPAN can be formulated as:

$$\max_{\mathcal{T}_1, \mathcal{T}_2 \in \mathcal{S}} \left\| \mathrm{eig}\left(\mathbf{L}_1\right) - \mathrm{eig}\left(\mathbf{L}_2\right) \right\|_2^2 \tag{8}$$

where the transformations $\mathcal{T}_1$ and $\mathcal{T}_2$ convert $\mathbf{A}$ to $\mathbf{A}_1$ and $\mathbf{A}_2$, respectively, producing the normalized Laplacian matrices $\mathbf{L}_1$ and $\mathbf{L}_2$. Here, $\mathcal{S}$ represents the set of all possible transformations, and the graph spectrum can be calculated by $\mathrm{eig}\left(\mathbf{L}\right)$.

## C Objective function of GCL framework

Here we briefly introduce the objective functions of the four **CG-SSL** frameworks used in this paper, for a more detailed discussion about objective functions including other graph contrastive learning and graph self-supervised learning frameworks which can refer to the survey papers (Xie et al., 2022; Wu et al., 2021; Liu et al., 2022b). We use the following notations:

- $p_\phi$: Projection head parameterized by $\phi$.

- $\mathbf{h}_i$, $\mathbf{h}_j$: Representations of the graph nodes.

- $\mathbf{h}'_n$: Representations of negative sample nodes.

- $\mathcal{P}$: Distribution of positive sample pairs.

- $\widetilde{\mathcal{P}}^N$: Distribution of negative sample pairs.

- $\mathcal{B}$: Set of nodes in a batch.

- $\mathbf{H}^{(1)}$, $\mathbf{H}^{(2)}$: Node representation matrices of two views.

GRACE uses the InfoNCE loss to maximize the similarity between positive pairs and minimize the similarity between negative pairs. InfoNCE loss encourages representations of positive pairs (generated from the same node via data augmentation) to be similar while pushing apart the representations of negative pairs (from different nodes). The loss function $\mathcal{L}_{\mathrm{NCE}}$ denotes as:

$$\mathcal{L}_{\mathrm{NCE}} \left(p_\phi \left(\mathbf{h}_i, \mathbf{h}_j\right)\right) = -\mathbb{E}_{\mathcal{P} \times \widetilde{\mathcal{P}}^N} \left[\log \frac{e^{p_\phi(\mathbf{h}_i, \mathbf{h}_j)}}{e^{p_\phi(\mathbf{h}_i, \mathbf{h}_j)} + \sum_{n \in N} e^{p_\phi(\mathbf{h}_i, \mathbf{h}'_n)}}\right] \tag{9}$$

MVGRL employs the Jensen-Shannon Estimator (JSE) for contrastive learning, which focuses on the mutual information between positive pairs and negative pairs. JSE maximizes the mutual information between positive pairs and minimizes it for negative pairs, thus improving the representations' alignment and uniformity. The loss function $\mathcal{L}_{\mathrm{JSE}}$ denotes as:

$$\mathcal{L}_{\mathrm{JSE}} \left(p_\phi \left(\mathbf{h}_i, \mathbf{h}_j\right)\right) = \mathbb{E}_{\mathcal{P} \times \tilde{\mathcal{P}}} \left[\log \left(1 - p_\phi \left(\mathbf{h}_i, \mathbf{h}'_j\right)\right)\right] - \mathbb{E}_{\mathcal{P}} \left[\log \left(p_\phi \left(\mathbf{h}_i, \mathbf{h}_j\right)\right)\right] \tag{10}$$

BGRL utilizes a loss similar to BYOL, which does not require negative samples. It uses two networks, an online network and a target network, to predict one view from the other:

$$\mathcal{L}_{\mathrm{BYOL}} \left(p_\phi \left(\mathbf{h}_i, \mathbf{h}_j\right)\right) = \mathbb{E}_{\mathcal{P} \times \mathcal{P}} \left[2 - 2 \cdot \frac{[p_\phi \left(\mathbf{h}_i\right)]^T \mathbf{h}_j}{\|p_\phi \left(\mathbf{h}_i\right)\| \, \|\mathbf{h}_j\|}\right] \tag{11}$$

G-BT applies the Barlow Twins' loss to reduce redundancy in the learned representations, thereby ensuring better generalization:

$$\begin{aligned}
\mathcal{L}_{\mathrm{BT}} \left(\mathbf{H}^{(1)}, \mathbf{H}^{(2)}\right) =& \mathbb{E}_{\mathcal{B} \sim \mathcal{P}|\mathcal{B}|} \left[\sum_a \left(1 - \frac{\sum_{i \in \mathcal{B}} \mathbf{H}_{ia}^{(1)} \mathbf{H}_{ia}^{(2)}}{\left\|\mathbf{H}_{ia}^{(1)}\right\| \left\|\mathbf{H}_{ia}^{(2)}\right\|}\right)^2 \right. \\
&\left. + \lambda \sum_a \sum_{b \neq a} \left(\frac{\sum_{i \in \mathcal{B}} \mathbf{H}_{ia}^{(1)} \mathbf{H}_{ib}^{(2)}}{\left\|\mathbf{H}_{ia}^{(1)}\right\| \left\|\mathbf{H}_{ib}^{(2)}\right\|}\right)^2 \right].
\end{aligned} \tag{12}$$

# D  Theoretical analysis

## D.1  Notations

Table 6: Notations and Definitions

| Notation | Definition |
|---|---|
| $\mathcal{G} = (\mathcal{V}, \mathcal{E})$ | Original undirected graph, where $\mathcal{V}$ is the set of nodes and $\mathcal{E}$ is the set of edges. |
| $\mathcal{G}'$ | Perturbed graph obtained from $\mathcal{G}$ via local topological perturbations. |
| $n = |\mathcal{V}|$ | Number of nodes in the graph. |
| $v \in \mathcal{V}$ | A node in the graph. |
| $\mathcal{G}_v^k$ | $k$-hop subgraph around node $v$ in $\mathcal{G}$. |
| $\mathcal{E}(\mathcal{G}_v^k)$ | Set of edges in the subgraph $\mathcal{G}_v^k$. |
| $|\mathcal{E}_v| = |\mathcal{E}(\mathcal{G}_v^k)|$ | Number of edges in the subgraph $\mathcal{G}_v^k$. |
| $n_v$ | Number of nodes in the subgraph $\mathcal{G}_v^k$. |
| $d_v, d_v'$ | Degrees of node $v$ in $\mathcal{G}$ and $\mathcal{G}'$, respectively. |
| $d_{\min}, d_{\max}$ | Minimum and maximum degrees in the $k$-hop subgraphs. |
| $\mathbf{A}, \mathbf{A}_v$ | Adjacency matrix of $\mathcal{G}$ and the adjacency matrix of $\mathcal{G}_v^k$, respectively. |
| $\mathbf{A}', \mathbf{A}'_v$ | Adjacency matrix of $\mathcal{G}'$ and the adjacency matrix of $\mathcal{G}'^k_v$, respectively. |
| $\mathbf{D}, \mathbf{D}_v$ | Degree matrix of $\mathcal{G}$ and the degree matrix of $\mathcal{G}_v^k$, respectively. |
| $\mathbf{D}', \mathbf{D}'_v$ | Degree matrix of $\mathcal{G}'$ and the degree matrix of $\mathcal{G}'^k_v$, respectively. |
| $\tilde{\mathbf{A}}'$ and $\tilde{\mathbf{A}}'_v$ | Normalized adjacency matrices of $\mathcal{G}'$ and $\mathcal{G}'^k_v$, respectively. |
| $\mathbf{X} \in \mathbb{R}^{n \times d_0}$ | Node feature matrix, where $d_0$ is the input feature dimension. |
| $k$ | Number of layers in the GNN and the size of the $k$-hop neighborhood. |
| $\mathbf{H}^{(l)} \in \mathbb{R}^{n \times d_l}$ | Hidden representations at layer $l$ in the GNN. |
| $\mathbf{W}^{(l)} \in \mathbb{R}^{d_{l-1} \times d_l}$ | Weight matrix at layer $l$ in the GNN, with $\left\|\mathbf{W}^{(l)}\right\|_2 \leq L_W$. |
| $L_W$ | Upper bound on the spectral norm of the weight matrices. |
| $\mathbf{h}_v \in \mathbb{R}^{d_k}$ | Embedding of node $v$ after $k$ GNN layers in $\mathcal{G}$. |
| $\mathbf{h}'_v \in \mathbb{R}^{d_k}$ | Embedding of node $v$ after $k$ GNN layers in $\mathcal{G}'$. |
| $\mathbf{P} \in \mathbb{R}^{d_k \times d}$ | Projection matrix applied to node embeddings to obtain final representations. |
| $\mathbf{z}_v = \mathbf{P}\mathbf{h}_v \in \mathbb{R}^d$ | Final embedding of node $v$ in $\mathcal{G}$ after projection. |
| $\mathbf{z}'_v = \mathbf{P}\mathbf{h}'_v \in \mathbb{R}^d$ | Final embedding of node $v$ in $\mathcal{G}'$ after projection. |
| $d$ | Embedding dimension of the final node representations. |
| $\tau$ | Temperature parameter in the InfoNCE loss. |
| $\mathrm{sim}(\mathbf{u}, \mathbf{v})$ | Cosine similarity between vectors $\mathbf{u}$ and $\mathbf{v}$, defined as $\mathrm{sim}(\mathbf{u}, \mathbf{v}) = \frac{\mathbf{u}^\top \mathbf{v}}{\|\mathbf{u}\|\|\mathbf{v}\|}$. |

## D.2  Definitions and Preliminaries

**Definition 1** (Local Topological Perturbation). *For a $k$-layer GNN, the* local topological perturbation *strength $\delta$ is defined as the maximum fraction of edge changes in any node's $k$-hop neighborhood:*

$$\delta = \max_{v \in \mathcal{V}} \frac{|\mathcal{E}(\mathcal{G}_v^k) \triangle \mathcal{E}(\mathcal{G}'^k_v)|}{|\mathcal{E}(\mathcal{G}_v^k)|}, \tag{1}$$

*where $\triangle$ denotes the symmetric difference of edge sets, and $\mathcal{G}'$ is the perturbed graph.*

**Definition 2** (InfoNCE Loss). *For a pair of graphs $(\mathcal{G}, \mathcal{G}')$, the* InfoNCE loss *is defined as:*

$$\mathcal{L}_{InfoNCE}(\mathcal{G}, \mathcal{G}') = -\frac{1}{n} \sum_{v \in \mathcal{V}} \log \frac{\exp\left(sim\left(\mathbf{z}_v, \mathbf{z}'_v\right)/\tau\right)}{\sum_{u \in \mathcal{V}} \exp\left(sim\left(\mathbf{z}_v, \mathbf{z}'_u\right)/\tau\right)} \tag{2}$$

*where $\mathbf{z}_v$ and $\mathbf{z}'_v$ are embeddings of node $v$ in $\mathcal{G}$ and $\mathcal{G}'$ respectively, $sim(\cdot, \cdot)$ is cosine similarity, $\tau$ is a temperature parameter, and $n = |\mathcal{V}|$.*

### D.3 Lemmas

**Lemma 1** (Adjacency Matrix Perturbation). *Given perturbation strength $\delta$, the change in adjacency matrices of the k-hop subgraph around any node $v$ satisfies:*

$$\|\mathbf{A}_v - \mathbf{A}'_v\|_F \leq \sqrt{2\delta|\mathcal{E}_v|}, \tag{3}$$

*where $\mathcal{G}_v^k$ denote the k-hop subgraph around node $v$ in the original graph $\mathcal{G}$, with adjacency matrix $\mathbf{A}_v$ and degree matrix $\mathbf{D}_v$. $|\mathcal{E}_v|$ is the number of edges in the k-hop subgraph $\mathcal{G}_v^k$. Similar notations for $\mathcal{G}_v^{k'}$, too.*

*Proof.* Each edge change affects two symmetric entries in the adjacency matrix $\mathbf{A}_v - \mathbf{A}'_v$, each with magnitude 1 (since edges are undirected). Let $m$ be the number of edge changes within $\mathcal{G}_v^k$. Then the Frobenius norm of the difference is:

$$\|\mathbf{A}_v - \mathbf{A}'_v\|_F^2 = \sum_{i,j} |A_{v,ij} - A'_{v,ij}|^2 = 2m. \tag{4}$$

Since the number of edge changes $m \leq \delta|\mathcal{E}_v|$, we have:

$$\|\mathbf{A}_v - \mathbf{A}'_v\|_F \leq \sqrt{2\delta|\mathcal{E}_v|}. \tag{5}$$

$\square$

**Lemma 2** (Degree Matrix Change). *For any node $v$ in $\mathcal{G}_v^k$:*

$$|d_v - d'_v| \leq \delta d_v. \tag{6}$$

*Moreover, for the degree matrices:*

$$\left\|\mathbf{D}_v^{-1/2} - \mathbf{D}'_v{}^{-1/2}\right\|_F \leq \frac{\delta\sqrt{n_v}}{2\sqrt{d_{\min}}(1-\delta)^{3/2}}, \tag{7}$$

*and*

$$\left\|\mathbf{D}_v^{-1/2} - \mathbf{D}'_v{}^{-1/2}\right\|_2 \leq \frac{\delta}{2\sqrt{d_{\min}}(1-\delta)^{3/2}}, \tag{8}$$

*where $n_v$ is the number of nodes in the k-hop subgraph, and $d_{\min}$ is the minimum degree in the subgraph.*

*Proof.* The degree of a node $v$ changes by at most $\delta d_v$ due to the perturbation:

$$|d_v - d'_v| \leq \delta d_v. \tag{9}$$

Consider the function $f(x) = x^{-1/2}$, which is convex for $x > 0$. Using the mean value theorem, for some $\xi_v$ between $d_v$ and $d'_v$:

$$d_v^{-1/2} - d'_v{}^{-1/2} = f'(\xi_v)(d_v - d'_v) = -\frac{1}{2}\xi_v^{-3/2}(d_v - d'_v). \tag{10}$$

Since $d'_v \geq (1-\delta)d_v$, we have $\xi_v \geq (1-\delta)d_v \geq (1-\delta)d_{\min}$. Thus,

$$\left|d_v^{-1/2} - d'_v{}^{-1/2}\right| \leq \frac{\delta d_v}{2((1-\delta)d_{\min})^{3/2}} = \frac{\delta d_v}{2(1-\delta)^{3/2}d_{\min}^{3/2}}. \tag{11}$$

Since $d_v \leq d_{\max}$, and $d_{\min} \leq d_v$, we have:

$$\left|d_v^{-1/2} - d'_v{}^{-1/2}\right| \leq \frac{\delta d_{\max}}{2(1-\delta)^{3/2}d_{\min}^{3/2}} \leq \frac{\delta}{2\sqrt{d_{\min}}(1-\delta)^{3/2}}. \tag{12}$$

The Frobenius norm is computed as:

$$\left\|\mathbf{D}_v^{-1/2} - \mathbf{D}'_v{}^{-1/2}\right\|_F^2 = \sum_v \left|d_v^{-1/2} - d'_v{}^{-1/2}\right|^2 \leq n_v\left(\frac{\delta}{2\sqrt{d_{\min}}(1-\delta)^{3/2}}\right)^2. \tag{13}$$

Therefore,

$$\left\| \mathbf{D}_v^{-1/2} - \mathbf{D}_v'^{-1/2} \right\|_F \leq \frac{\delta \sqrt{n_v}}{2\sqrt{d_{\min}}(1-\delta)^{3/2}}. \tag{14}$$

Similarly, the spectral norm bound is:

$$\left\| \mathbf{D}_v^{-1/2} - \mathbf{D}_v'^{-1/2} \right\|_2 \leq \frac{\delta}{2\sqrt{d_{\min}}(1-\delta)^{3/2}}. \tag{15}$$

$\square$

**Lemma 3** (Bounded Change in Normalized Adjacency Matrix). *Given a graph $\mathcal{G}$ with minimum degree $d_{\min}$, maximum degree $d_{\max}$, and $n_v$ nodes in the k-hop subgraph, and its perturbation $\mathcal{G}'$ with local topological perturbation strength $\delta$, the change in the normalized adjacency matrix for any k-hop subgraph is bounded by:*

$$\left\| \tilde{\mathbf{A}}_v - \tilde{\mathbf{A}}_v' \right\|_F \leq \frac{\sqrt{n_v d_{\max}}}{d_{\min}} \left( \sqrt{\delta} + \frac{\delta}{(1-\delta)^{3/2}} \right). \tag{16}$$

*Proof.* We start by noting that the normalized adjacency matrix is given by $\tilde{\mathbf{A}}_v = \mathbf{D}_v^{-1/2} \mathbf{A}_v \mathbf{D}_v^{-1/2}$. The difference between the normalized adjacency matrices is:

$$\tilde{\mathbf{A}}_v - \tilde{\mathbf{A}}_v' = \mathbf{D}_v^{-1/2} \mathbf{A}_v \mathbf{D}_v^{-1/2} - \mathbf{D}_v'^{-1/2} \mathbf{A}_v' \mathbf{D}_v'^{-1/2}. \tag{17}$$

Add and subtract $\mathbf{D}_v^{-1/2} \mathbf{A}_v' \mathbf{D}_v^{-1/2}$:

$$\tilde{\mathbf{A}}_v - \tilde{\mathbf{A}}_v' = \mathbf{D}_v^{-1/2} (\mathbf{A}_v - \mathbf{A}_v') \mathbf{D}_v^{-1/2} + (\mathbf{D}_v^{-1/2} \mathbf{A}_v' \mathbf{D}_v^{-1/2} - \mathbf{D}_v'^{-1/2} \mathbf{A}_v' \mathbf{D}_v'^{-1/2}). \tag{18}$$

Let $\mathbf{E} = \mathbf{D}_v^{-1/2} \mathbf{A}_v' \mathbf{D}_v^{-1/2} - \mathbf{D}_v'^{-1/2} \mathbf{A}_v' \mathbf{D}_v'^{-1/2}$. Then,

$$\tilde{\mathbf{A}}_v - \tilde{\mathbf{A}}_v' = \mathbf{D}_v^{-1/2} (\mathbf{A}_v - \mathbf{A}_v') \mathbf{D}_v^{-1/2} + \mathbf{E}. \tag{19}$$

First, we bound the first term:

$$\left\| \mathbf{D}_v^{-1/2} (\mathbf{A}_v - \mathbf{A}_v') \mathbf{D}_v^{-1/2} \right\|_F \leq \left\| \mathbf{D}_v^{-1/2} \right\|_2^2 \left\| \mathbf{A}_v - \mathbf{A}_v' \right\|_F \leq \frac{1}{d_{\min}} \sqrt{2\delta |\mathcal{E}_v|}. \tag{20}$$

Since $|\mathcal{E}_v| \leq \frac{1}{2} n_v d_{\max}$, we have:

$$\sqrt{2\delta |\mathcal{E}_v|} \leq \sqrt{2\delta \left( \frac{1}{2} n_v d_{\max} \right)} = \sqrt{\delta n_v d_{\max}}. \tag{21}$$

Thus,

$$\left\| \mathbf{D}_v^{-1/2} (\mathbf{A}_v - \mathbf{A}_v') \mathbf{D}_v^{-1/2} \right\|_F \leq \frac{\sqrt{\delta n_v d_{\max}}}{d_{\min}}. \tag{22}$$

Next, we bound $\|\mathbf{E}\|_F$. Note that:

$$\mathbf{E} = (\mathbf{D}_v^{-1/2} - \mathbf{D}_v'^{-1/2}) \mathbf{A}_v' \mathbf{D}_v^{-1/2} + \mathbf{D}_v'^{-1/2} \mathbf{A}_v' (\mathbf{D}_v^{-1/2} - \mathbf{D}_v'^{-1/2}). \tag{23}$$

Therefore,

$$\|\mathbf{E}\|_F \leq 2 \left\| \mathbf{D}_v^{-1/2} - \mathbf{D}_v'^{-1/2} \right\|_2 \|\mathbf{A}_v'\|_F \left\| \mathbf{D}_v^{-1/2} \right\|_2. \tag{24}$$

Since $\|\mathbf{A}_v'\|_F \leq \sqrt{2|\mathcal{E}_v|} \leq \sqrt{n_v d_{\max}}$, $\left\| \mathbf{D}_v^{-1/2} \right\|_2 \leq \frac{1}{\sqrt{d_{\min}}}$, and using the bound from Lemma 2 for $\left\| \mathbf{D}_v^{-1/2} - \mathbf{D}_v'^{-1/2} \right\|_2$, we have:

$$\|\mathbf{E}\|_F \leq 2 \times \frac{\delta}{2\sqrt{d_{\min}}(1-\delta)^{3/2}} \times \sqrt{n_v d_{\max}} \times \frac{1}{\sqrt{d_{\min}}} = \frac{\delta \sqrt{n_v d_{\max}}}{d_{\min}(1-\delta)^{3/2}}. \tag{25}$$

Combining both terms:

$$\left\|\tilde{\mathbf{A}}_v - \tilde{\mathbf{A}}'_v\right\|_F \leq \frac{\sqrt{\delta n_v d_{\max}}}{d_{\min}} + \frac{\delta\sqrt{n_v d_{\max}}}{d_{\min}(1-\delta)^{3/2}} = \frac{\sqrt{n_v d_{\max}}}{d_{\min}}\left(\sqrt{\delta} + \frac{\delta}{(1-\delta)^{3/2}}\right). \tag{26}$$

$\square$

**Lemma 4** (GNN Output Difference Bound)**.** *For a k-layer Graph Neural Network (GNN) $f_\theta$ with ReLU activation functions and weight matrices satisfying $\left\|\mathbf{W}^{(l)}\right\|_2 \leq L_W$ for all layers l, given two graphs $\mathcal{G}$ and $\mathcal{G}'$ with local topological perturbation strength $\delta$, the difference in GNN outputs for any node v is bounded by:*

$$\|\mathbf{h}_v - \mathbf{h}'_v\| \leq k\left(AL_W\right)^k B\|\mathbf{X}\|_2, \tag{27}$$

*where:*

- $\mathbf{h}_v$ *and* $\mathbf{h}'_v$ *are the embeddings of node v in* $\mathcal{G}$ *and* $\mathcal{G}'$*, respectively, after k GNN layers.*

- $\mathbf{X}$ *is the node feature matrix.*

- $A = \dfrac{\sqrt{n_v d_{\max}}}{d_{\min}}.$

- $B = \sqrt{\delta} + \dfrac{\delta}{(1-\delta)^{3/2}}.$

- $n_v$ *is the number of nodes in the k-hop subgraph around node v.*

- $d_{\min}$ *and* $d_{\max}$ *are the minimum and maximum degrees in the subgraph.*

*Proof.* We will prove the lemma by induction on the number of layers $l$.

**Base Case** ($l = 0$)**.** At layer $l = 0$, before any GNN layers are applied, the embeddings are simply the input features:

$$\mathbf{H}^{(0)} = \mathbf{X}, \quad \mathbf{H}'^{(0)} = \mathbf{X}. \tag{28}$$

Thus,

$$\left\|\mathbf{H}^{(0)} - \mathbf{H}'^{(0)}\right\|_F = 0. \tag{29}$$

This establishes the base case.

**Inductive Step.** Assume that for some $l \geq 0$, the following bound holds:

$$\left\|\mathbf{H}^{(l)} - \mathbf{H}'^{(l)}\right\|_F \leq l\left(AL_W\right)^l B\|\mathbf{X}\|_2. \tag{30}$$

Our goal is to show that the bound holds for layer $l + 1$:

$$\left\|\mathbf{H}^{(l+1)} - \mathbf{H}'^{(l+1)}\right\|_F \leq (l+1)\left(AL_W\right)^{l+1} B\|\mathbf{X}\|_2. \tag{31}$$

The outputs at layer $(l + 1)$ are:

$$\mathbf{H}^{(l+1)} = \text{ReLU}\left(\tilde{\mathbf{A}}\mathbf{H}^{(l)}\mathbf{W}^{(l)}\right), \quad \mathbf{H}'^{(l+1)} = \text{ReLU}\left(\tilde{\mathbf{A}}'\mathbf{H}'^{(l)}\mathbf{W}^{(l)}\right), \tag{32}$$

where:

- $\tilde{\mathbf{A}}$ and $\tilde{\mathbf{A}}'$ are the normalized adjacency matrices of the $k$-hop subgraphs around node $v$ in $\mathcal{G}$ and $\mathcal{G}'$, respectively.

- $\mathbf{W}^{(l)}$ is the weight matrix of layer $l$, with $\left\|\mathbf{W}^{(l)}\right\|_2 \leq L_W$.

Since ReLU is 1-Lipschitz, we have:

$$\left\|\mathbf{H}^{(l+1)} - \mathbf{H}'^{(l+1)}\right\|_F \leq \left\|\tilde{\mathbf{A}}\mathbf{H}^{(l)}\mathbf{W}^{(l)} - \tilde{\mathbf{A}}'\mathbf{H}'^{(l)}\mathbf{W}^{(l)}\right\|_F. \tag{33}$$

We can expand the difference as:

$$\tilde{\mathbf{A}}\mathbf{H}^{(l)}\mathbf{W}^{(l)} - \tilde{\mathbf{A}}'\mathbf{H}'^{(l)}\mathbf{W}^{(l)} = \underbrace{\tilde{\mathbf{A}}\left(\mathbf{H}^{(l)} - \mathbf{H}'^{(l)}\right)\mathbf{W}^{(l)}}_{T_1} + \underbrace{\left(\tilde{\mathbf{A}} - \tilde{\mathbf{A}}'\right)\mathbf{H}'^{(l)}\mathbf{W}^{(l)}}_{T_2}. \tag{34}$$

**Bounding $T_1$.** Using the submultiplicative property of norms:

$$\|T_1\|_F \leq \left\|\tilde{\mathbf{A}}\right\|_F \left\|\mathbf{H}^{(l)} - \mathbf{H}'^{(l)}\right\|_2 \left\|\mathbf{W}^{(l)}\right\|_2. \tag{35}$$

From properties of $\tilde{\mathbf{A}}$ and $\mathbf{W}^{(l)}$:

$$\left\|\tilde{\mathbf{A}}\right\|_F \leq A, \quad \text{(as shown below)}, \quad \left\|\mathbf{W}^{(l)}\right\|_2 \leq L_W. \tag{36}$$

Also, since $\left\|\mathbf{H}^{(l)} - \mathbf{H}'^{(l)}\right\|_2 \leq \left\|\mathbf{H}^{(l)} - \mathbf{H}'^{(l)}\right\|_F$, we have:

$$\|T_1\|_F \leq AL_W \left\|\mathbf{H}^{(l)} - \mathbf{H}'^{(l)}\right\|_F. \tag{37}$$

**Bounding $\left\|\tilde{\mathbf{A}}\right\|_F$.** The entries of $\tilde{\mathbf{A}}$ are:

$$\tilde{A}_{ij} = \frac{A_{ij}}{\sqrt{d_i d_j}}, \tag{38}$$

where $A_{ij} \in \{0, 1\}$, and $d_i, d_j \geq d_{\min}$. Therefore,

$$|\tilde{A}_{ij}| \leq \frac{1}{d_{\min}}. \tag{39}$$

The number of non-zero entries in $\tilde{\mathbf{A}}$ is at most $n_v d_{\max}$. Therefore,

$$\left\|\tilde{\mathbf{A}}\right\|_F \leq \frac{\sqrt{n_v d_{\max}}}{d_{\min}} = A. \tag{40}$$

**Bounding $T_2$.** Similarly, we have:

$$\|T_2\|_F \leq \left\|\tilde{\mathbf{A}} - \tilde{\mathbf{A}}'\right\|_F \left\|\mathbf{H}'^{(l)}\right\|_2 \left\|\mathbf{W}^{(l)}\right\|_2. \tag{41}$$

From the perturbation analysis:

$$\left\|\tilde{\mathbf{A}} - \tilde{\mathbf{A}}'\right\|_F \leq AB. \tag{42}$$

To bound $\left\|\mathbf{H}'^{(l)}\right\|_2$, we note that:

$$\left\|\mathbf{H}'^{(l)}\right\|_2 \leq \left\|\mathbf{H}'^{(l)}\right\|_F. \tag{43}$$

We can bound $\left\|\mathbf{H}'^{(l)}\right\|_F$ recursively.

**Bounding $\left\|\mathbf{H}'^{(l)}\right\|_F$.** At each layer, the output is given by:

$$\mathbf{H}'^{(l)} = \text{ReLU}\left(\tilde{\mathbf{A}}'\mathbf{H}'^{(l-1)}\mathbf{W}^{(l-1)}\right). \tag{44}$$

Since ReLU is 1-Lipschitz and $\left\|\tilde{\mathbf{A}}'\right\|_F \leq A$, we have:

$$\left\|\mathbf{H}'^{(l)}\right\|_F \leq \left\|\tilde{\mathbf{A}}'\mathbf{H}'^{(l-1)}\mathbf{W}^{(l-1)}\right\|_F \leq AL_W \left\|\mathbf{H}'^{(l-1)}\right\|_2. \tag{45}$$

Recursively applying this bound from $l = 0$ to $l$, and noting that $\left\|\mathbf{H}'^{(0)}\right\|_2 = \|\mathbf{X}\|_2$, we obtain:

$$\left\|\mathbf{H}'^{(l)}\right\|_F \leq (AL_W)^l \|\mathbf{X}\|_2. \tag{46}$$

Therefore,

$$\left\|\mathbf{H}'^{(l)}\right\|_2 \leq (AL_W)^l \|\mathbf{X}\|_2. \tag{47}$$

Now we have:

$$\|T_2\|_F \leq AB(AL_W)^l \|\mathbf{X}\|_2 L_W = A^{l+1}BL_W^{l+1} \|\mathbf{X}\|_2. \tag{48}$$

**Total Bound for $\left\|\mathbf{H}^{(l+1)} - \mathbf{H}'^{(l+1)}\right\|_F$.**

Combining $T_1$ and $T_2$:

$$\left\|\mathbf{H}^{(l+1)} - \mathbf{H}'^{(l+1)}\right\|_F \leq AL_W \left\|\mathbf{H}^{(l)} - \mathbf{H}'^{(l)}\right\|_F + A^{l+1}BL_W^{l+1} \|\mathbf{X}\|_2. \tag{49}$$

**Recursive Relation.** Let $C_l = \left\|\mathbf{H}^{(l)} - \mathbf{H}'^{(l)}\right\|_F$. The recursive relation is:

$$C_{l+1} \leq AL_W C_l + A^{l+1}BL_W^{l+1} \|\mathbf{X}\|_2. \tag{50}$$

We will prove by induction that:

$$C_l \leq l(AL_W)^l B \|\mathbf{X}\|_2. \tag{51}$$

**Base Case.** For $l = 0$, $C_0 = 0$, which satisfies the bound.

**Inductive Step.** Assume the bound holds for $l$:

$$C_l \leq l(AL_W)^l B \|\mathbf{X}\|_2. \tag{52}$$

Then for $l + 1$:

$$\begin{aligned}
C_{l+1} &\leq AL_W C_l + A^{l+1}BL_W^{l+1} \|\mathbf{X}\|_2 \\
&\leq AL_W \left(l(AL_W)^l B \|\mathbf{X}\|_2\right) + A^{l+1}BL_W^{l+1} \|\mathbf{X}\|_2 \\
&= lA^{l+1}L_W^{l+1} B \|\mathbf{X}\|_2 + A^{l+1}L_W^{l+1} B \|\mathbf{X}\|_2 \\
&= (l+1)A^{l+1}L_W^{l+1} B \|\mathbf{X}\|_2 \\
&= (l+1)(AL_W)^{l+1} B \|\mathbf{X}\|_2.
\end{aligned} \tag{53}$$

This confirms that the bound holds for $l + 1$.

For $l = k$, we have:

$$\left\|\mathbf{H}^{(k)} - \mathbf{H}'^{(k)}\right\|_F \leq k(AL_W)^k B \|\mathbf{X}\|_2. \tag{54}$$

Since $\|\mathbf{h}_v - \mathbf{h}'_v\| \leq \left\|\mathbf{H}^{(k)} - \mathbf{H}'^{(k)}\right\|_F$, we obtain:

$$\|\mathbf{h}_v - \mathbf{h}'_v\| \leq k \left(AL_W\right)^k B \left\|\mathbf{X}\right\|_2. \tag{55}$$

This completes the proof. $\qquad\square$

**Lemma 5** (Minimum Cosine Similarity for Positive Pairs). *For embeddings $\mathbf{z}_v$ and $\mathbf{z}'_v$ produced by a linear projection of GNN outputs, with $\|\mathbf{z}_v\| = \|\mathbf{z}'_v\| = 1$, the cosine similarity satisfies:*

$$sim\left(\mathbf{z}_v, \mathbf{z}'_v\right) \geq 1 - \frac{\epsilon^2}{2}, \tag{56}$$

*where*

$$\epsilon = \frac{k(\frac{\sqrt{n_v d_{\max}}}{d_{\min}})^k \left(\sqrt{\delta} + \frac{\delta}{(1-\delta)^{3/2}}\right) L_W^k \left\|\mathbf{X}\right\|_2 \left\|\mathbf{P}\right\|_2}{c_z}, \tag{57}$$

*and $c_z$ is the lower bound on $\|\mathbf{z}_v\|$ (which equals 1 in this case).*

*Proof.* The embeddings are computed as $\mathbf{z}_v = \mathbf{P}\mathbf{h}_v$ and $\mathbf{z}'_v = \mathbf{P}\mathbf{h}'_v$. Then,

$$\|\mathbf{z}_v - \mathbf{z}'_v\| \leq \left\|\mathbf{P}\right\|_2 \left\|\mathbf{h}_v - \mathbf{h}'_v\right\|. \tag{58}$$

Using the bound from Lemma 4, we have:

$$\|\mathbf{z}_v - \mathbf{z}'_v\| \leq \left(\frac{\sqrt{n_v d_{\max}}}{d_{\min}} \left(\sqrt{\delta} + \frac{\delta}{(1-\delta)^{3/2}}\right) L_W^k \left\|\mathbf{X}\right\|_2 \left\|\mathbf{P}\right\|_2\right). \tag{59}$$

Since $\|\mathbf{z}_v\| = \|\mathbf{z}'_v\| = 1$, the cosine similarity satisfies:

$$\mathrm{sim}\left(\mathbf{z}_v, \mathbf{z}'_v\right) = 1 - \frac{1}{2} \left\|\mathbf{z}_v - \mathbf{z}'_v\right\|^2 \geq 1 - \frac{\epsilon^2}{2}. \tag{60}$$

$\qquad\square$

**Lemma 6** (Refined Negative Pair Similarity Bound). *Assuming that embeddings of different nodes are approximately independent and randomly oriented in high-dimensional space, and that the embedding dimension $d$ satisfies $d \gg \log n$, we have, with high probability:*

$$|sim(\mathbf{z}_v, \mathbf{z}'_u)| \leq \epsilon', \tag{61}$$

*where*

$$\epsilon' = \sqrt{\frac{2 \log n}{d}}. \tag{62}$$

*Proof.* Since $\mathbf{z}_v$ and $\mathbf{z}'_u$ are unit vectors in $\mathbb{R}^d$ and approximately independent for $u \neq v$, the inner product $\langle \mathbf{z}_v, \mathbf{z}'_u \rangle$ follows a distribution with mean zero and variance $\frac{1}{d}$. By applying concentration inequalities such as Hoeffding's inequality or the Gaussian tail bound, for any $\epsilon' > 0$:

$$P\left(|\langle \mathbf{z}_v, \mathbf{z}'_u \rangle| \geq \epsilon'\right) \leq 2\exp\left(-\frac{d(\epsilon')^2}{2}\right). \tag{63}$$

Selecting $\epsilon' = \sqrt{\frac{2 \log n}{d}}$, we get:

$$P\left(|\langle \mathbf{z}_v, \mathbf{z}'_u \rangle| \geq \sqrt{\frac{2 \log n}{d}}\right) \leq \frac{2}{n}. \tag{64}$$

Using the union bound over all $n(n-1)$ pairs, the probability that any pair violates this bound is small when $d \gg \log n$. $\qquad\square$

### D.4 Main Theorem

**Theorem 1** (InfoNCE Loss Bounds). *Given a graph $\mathcal{G}$ with minimum degree $d_{\min}$ and maximum degree $d_{\max}$, and its augmentation $\mathcal{G}'$ with local topological perturbation strength $\delta$, for a k-layer GNN with ReLU activation and weight matrices satisfying $\left\|\mathbf{W}^{(l)}\right\|_2 \leq L_W$, and assuming that the embeddings are normalized ($\|\mathbf{z}_v\| = \|\mathbf{z}_v'\| = 1$), the InfoNCE loss satisfies with high probability:*

$$-\log\left(\frac{e^{1/\tau}}{e^{1/\tau} + (n-1)e^{-\epsilon'/\tau}}\right) \leq \mathcal{L}_{InfoNCE}(\mathcal{G}, \mathcal{G}') \leq -\log\left(\frac{e^{\left(1-\frac{\epsilon^2}{2}\right)/\tau}}{e^{\left(1-\frac{\epsilon^2}{2}\right)/\tau} + (n-1)e^{\epsilon'/\tau}}\right), \tag{65}$$

*where $\epsilon$ is as defined in Lemma 5 and $\epsilon'$ is as defined in Lemma 6.*

*Proof.* For the positive pairs (same node in $\mathcal{G}$ and $\mathcal{G}'$), from Lemma 5:

$$\text{sim}\left(\mathbf{z}_v, \mathbf{z}_v'\right) \geq 1 - \frac{\epsilon^2}{2}. \tag{66}$$

For the negative pairs (different nodes), from Lemma 6, with high probability:

$$\left|\text{sim}\left(\mathbf{z}_v, \mathbf{z}_u'\right)\right| \leq \epsilon', \quad \forall u \neq v. \tag{67}$$

The InfoNCE loss for node $v$ is:

$$\mathcal{L}_v = -\log \frac{\exp\left(\text{sim}\left(\mathbf{z}_v, \mathbf{z}_v'\right)/\tau\right)}{\exp\left(\text{sim}\left(\mathbf{z}_v, \mathbf{z}_v'\right)/\tau\right) + \sum_{u \neq v} \exp\left(\text{sim}\left(\mathbf{z}_v, \mathbf{z}_u'\right)/\tau\right)}. \tag{68}$$

For the **upper bound** on $\mathcal{L}_v$, we use the minimal positive similarity and maximal negative similarity:

$$\mathcal{L}_v \leq -\log \frac{e^{\left(1-\frac{\epsilon^2}{2}\right)/\tau}}{e^{\left(1-\frac{\epsilon^2}{2}\right)/\tau} + (n-1)e^{\epsilon'/\tau}}. \tag{69}$$

For the **lower bound** on $\mathcal{L}_v$, we use the maximal positive similarity and minimal negative similarity:

$$\mathcal{L}_v \geq -\log \frac{e^{1/\tau}}{e^{1/\tau} + (n-1)e^{-\epsilon'/\tau}}. \tag{70}$$

Since this holds for all nodes $v$, averaging over all nodes, we obtain the bounds for $\mathcal{L}_{\text{InfoNCE}}(\mathcal{G}, \mathcal{G}')$. $\qquad\square$

### D.5 Numerical Estimation

To assess how tight the bound is while keeping $d_{\min}$ not too large (e.g., $d_{\min} = 10$), let's perform a numerical estimation.

Suppose:

- Number of nodes: $n = 1000$.

- Embedding dimension: $d = 4096$.

- Minimum degree: $d_{\min} = 10$.

- Maximum degree: $d_{\max} = 30$.

- Layer count: $k = 1$.

- Weight matrix norm: $L_W = 0.5$.

- Input feature norm: $\|\mathbf{X}\|_2 = 1$.

- Projection matrix norm: $\|\mathbf{P}\|_2 = 1$.

- Temperature: $\tau = 0.5$.

- Local perturbation strength: $\delta = 0.1$.

Compute $\epsilon$:

Assuming $n_v \approx 30$,

$$
\begin{aligned}
\sqrt{n_v d_{\max}} &= \sqrt{30 \times 30} = \sqrt{900} = 30, \\
\frac{\sqrt{n_v d_{\max}}}{d_{\min}} &= \frac{30}{10} = 3, \\
\sqrt{\delta} &= \sqrt{0.1} \approx 0.316228, \\
\frac{\delta}{(1-\delta)^{3/2}} &\approx \frac{0.1}{(1-0.1)^{1.5}} \approx \frac{0.1}{0.853814} \approx 0.117121, \\
\sqrt{\delta} + \frac{\delta}{(1-\delta)^{3/2}} &\approx 0.316228 + 0.117121 = 0.433349, \\
\epsilon &= 1 \times 3 \times 0.433349 \times 0.5 \times 1 \times 1 \approx 0.650.
\end{aligned}
$$

Compute $\epsilon'$:

$$
\epsilon' = \sqrt{\frac{2 \log n}{d}} = \sqrt{\frac{13.8155}{4096}} \approx \sqrt{0.003374} \approx 0.05805.
$$

Compute exponents:

For the upper bound:

$$
\frac{1 - \frac{\epsilon^2}{2}}{\tau} = 1.5775, \quad \frac{\epsilon'}{\tau} = \frac{0.05805}{0.5} = 0.1161.
$$

For the lower bound:

$$
\frac{1}{\tau} = 2, \quad \frac{-\epsilon'}{\tau} = -0.1161.
$$

Compute numerator and denominator for the upper bound:

$$
\begin{aligned}
\text{Numerator} &\approx 4.8426, \\
\text{Denominator} &\approx 1126.8881.
\end{aligned}
$$

Compute numerator and denominator for the lower bound:

$$\text{Numerator} \approx 7.3891,$$
$$\text{Denominator} \approx 896.5990.$$

Compute the InfoNCE loss bounds:

$$\mathcal{L}_{\text{upper}} = -\log\left(\frac{4.8426}{1126.8881}\right) = -\log(0.003964) \approx 5.4497,$$
$$\mathcal{L}_{\text{lower}} = -\log\left(\frac{7.3891}{896.5990}\right) = -\log(0.00824) \approx 4.7989.$$

**Interpretation.** The numerical gap between the upper and lower bounds, calculated as $5.4497 - 4.7989 = 0.6508$, is notably narrow. This tight interval highlights a key observation: shallow GNNs face intrinsic challenges in effectively exploiting spectral enhancement techniques. This is due to their restricted capacity to represent and process the spectral characteristics of a graph, irrespective of the complexity of the spectral modifications applied. The findings suggest that tuning fundamental augmentation parameters, such as perturbation strength, may exert a more pronounced influence on learning outcomes than intricate spectral alterations. While the theoretical rationale behind spectral augmentations is well-motivated, their practical utility might only be realized when paired with deeper GNNs capable of leveraging augmented spectral information across multiple layers of message propagation.

### D.6 Discussions on Theorem 1

In this section, we generally describe the claim we want to conclude from Theorem 1 and explain its connection to spectral augmentations.

#### D.6.1 Positioning of Theorem 1

In general, Theorem 1 offers a useful lens by showing that under common assumptions, the InfoNCE objective remains bounded within a finite range regarding the magnitude of local topological perturbation. We also carry out empirical validation of our theoretical InfoNCE bounds in Appendix D.6. However, it should be noted that it is indirect to some extent to spectral augmentations, due to the inherent difficulty of providing direct theoretical proof that simultaneously addresses spectral information, InfoNCE training objective, and downstream tasks in CG-SSL.

#### D.6.2 Connections to Spectral Augmentations

Because spectral augmentations primarily target the graph's eigenstructure, they may not necessarily push the loss beyond these inherent bounds, where the loss is critical to the optimization of the CG-SSL and eventually to the performance of downstream tasks. Therefore, we view Theorem 1 as an initial theoretical insight rather than a definitive statement on the value of spectral augmentations. By highlighting that the InfoNCE loss is constrained, it adds context to why advanced augmentation schemes and simpler approaches may converge to similar performance levels. We remain optimistic that further theoretical and empirical investigations can deepen our understanding of how spectral perturbations, and graph augmentations, interact with contrastive objectives in CG-SSL.

#### D.6.3 Coverage of Theorem 1

**Loss functions.** Theorem 1 specifically focuses on establishing bounds for the InfoNCE objective in shallow GNNs. We select InfoNCE due to its prominence and widespread adoption in contrastive graph self-supervised learning (CG-SSL). While this theoretical result provides valuable insights into the behavior of InfoNCE-based CG-SSL, it should be noted that the bounds may not directly generalize to other contrastive objectives. While potentially valuable, the analysis of other objectives falls outside the scope of our current work, which aims to establish a foundational theoretical understanding of this widely-used objective.

**Number of layers of GNN encoder.** Note that Theorem 1 is proposed specifically for shallow GNNs and does not claim to directly extend to deeper architectures. Nonetheless, our empirical results indicate that deeper GNNs can exhibit degraded performance in practice, which is consistent with real-world setups for spectral augmentation approaches as well (SPAN (Lin et al., 2023) uses 2 layers, SpCo (Liu et al., 2022a) has only 1 layer and GASSER (Yang et al., 2023) uses 1 or 2 layers). This outcome is not surprising, since shallow GNN encoders are widely adopted by various methods in the domain of CG-SSL, mostly for the sake of empirical performance. At the same time, we acknowledge that more work can be done to explore, especially from the theoretical aspect, how deeper GNN architectures might be effectively coupled with spectral augmentations, as the interactions in that setting are far more complex than those considered in our current theoretical result for shallow layers.

## D.7 Empirical Verification of InfoNCE Bounds

To empirically validate our theoretical bounds on the InfoNCE loss (Theorem 1) and investigate the necessity of spectral augmentations, we conduct extensive experiments across different datasets using GRACE with a two-layer GCN encoder in Fig. 5 and one-layer GCN encoder in Fig. 6. We fix the edge dropping rate at 0.3 and perform 100 independent runs with different random seeds. These multiple runs naturally induce various spectral changes through random edge perturbations, allowing us to examine how the InfoNCE loss behaves under a wide range of spectral variations.

Our experiments on MUTAG and PROTEINS datasets reveal that the InfoNCE loss exhibits remarkably stable behavior, with minimal variations across different runs. This stability persists despite the diverse spectral changes induced by random edge dropping, supporting our argument that complex spectral augmentations may be unnecessary as simple edge perturbations already provide sufficient regularization.

Interestingly, on the dense IMDB dataset, we observe larger fluctuations in the InfoNCE loss during the middle stages of training, with variations occasionally exceeding 1.0. However, these fluctuations consistently converge to a narrow band during the final stages of training, aligning with our theoretical predictions. It's worth noting that our experimental setup using a two-layer GCN with 1024-dimensional embeddings theoretically allows for wider bounds compared to the single-layer GCN with 4096-dimensional embeddings analyzed in our proof. The fact that the observed loss values remain well within our theoretical bounds suggests that our derivation provides conservative estimates of the actual constraints on the InfoNCE loss.

These empirical results strengthen our theoretical findings by demonstrating that even with the diverse spectral variations induced by random edge perturbations, the InfoNCE loss remains well-bounded. This suggests that explicit spectral augmentations may be redundant, as simple edge perturbations already provide effective regularization while being computationally more efficient.

# E More experiments

## E.1 Effect of numbers of GCN Layers

We explore the impact of GCN depth on accuracy by testing GCNs with 4, 6, and 8 layers, using our edge perturbation methods alongside SPAN baselines. Experiments were conducted with the GRACE and G-BT frameworks on the Cora dataset for node classification and the MUTAG dataset for graph classification. Each configuration was run three times, with the mean accuracy and standard deviation reported.

Overall, deeper GCNs (6 and 8 layers) tend to perform worse across both tasks, reinforcing the observation that deeper architectures, despite their theoretical expressive power, may negatively impact the quality of learned representations. The results are summarized in Tables 7 and 8.

## E.2 Effect of GNN encoder

To further validate the generality of our approach, we conducted additional experiments using different GNN encoders. For the node classification task, we evaluated the CORA dataset with GAT as the encoder, while

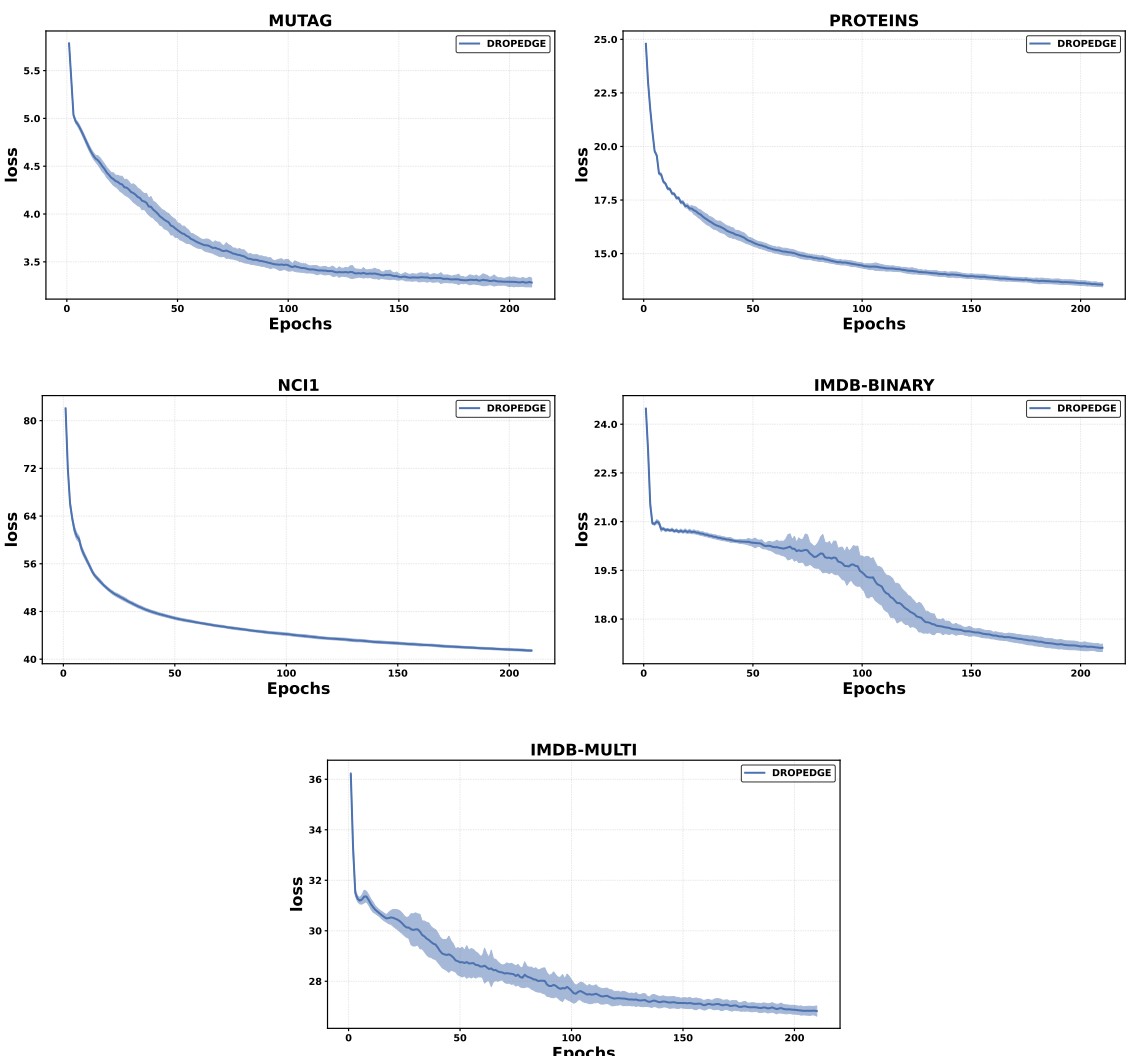

Figure 5: Empirical behavior of InfoNCE loss across different datasets with GRACE and two-layer GCN. We show the mean InfoNCE loss (solid blue line) and standard deviation (shaded area) over 100 independent runs with random edge perturbations (drop rate = 0.3). Results are presented for MUTAG, PROTEINS, IMDB-BINARY, IMDB-MULTI, and NCI1 datasets using a two-layer GCN encoder. The loss values exhibit consistent convergence to stable ranges, even though each run induces different spectral variations through random edge dropping. While the dense IMDB dataset shows larger fluctuations during the middle stages of training, all datasets eventually exhibit well-bounded behavior, suggesting that simple edge perturbations provide sufficient regularization for contrastive learning.

for the graph classification task, we performed experiments on the MUTAG dataset using both GAT and GPS as encoders.

The results, presented in Tables 9 and 10, are shown alongside the results obtained with GCN encoders. These findings demonstrate that our simple edge perturbation method consistently outperforms the baselines, regardless of the choice of the encoder. This confirms that our conclusions hold across different encoder architectures, underscoring the robustness and effectiveness of the proposed approach.

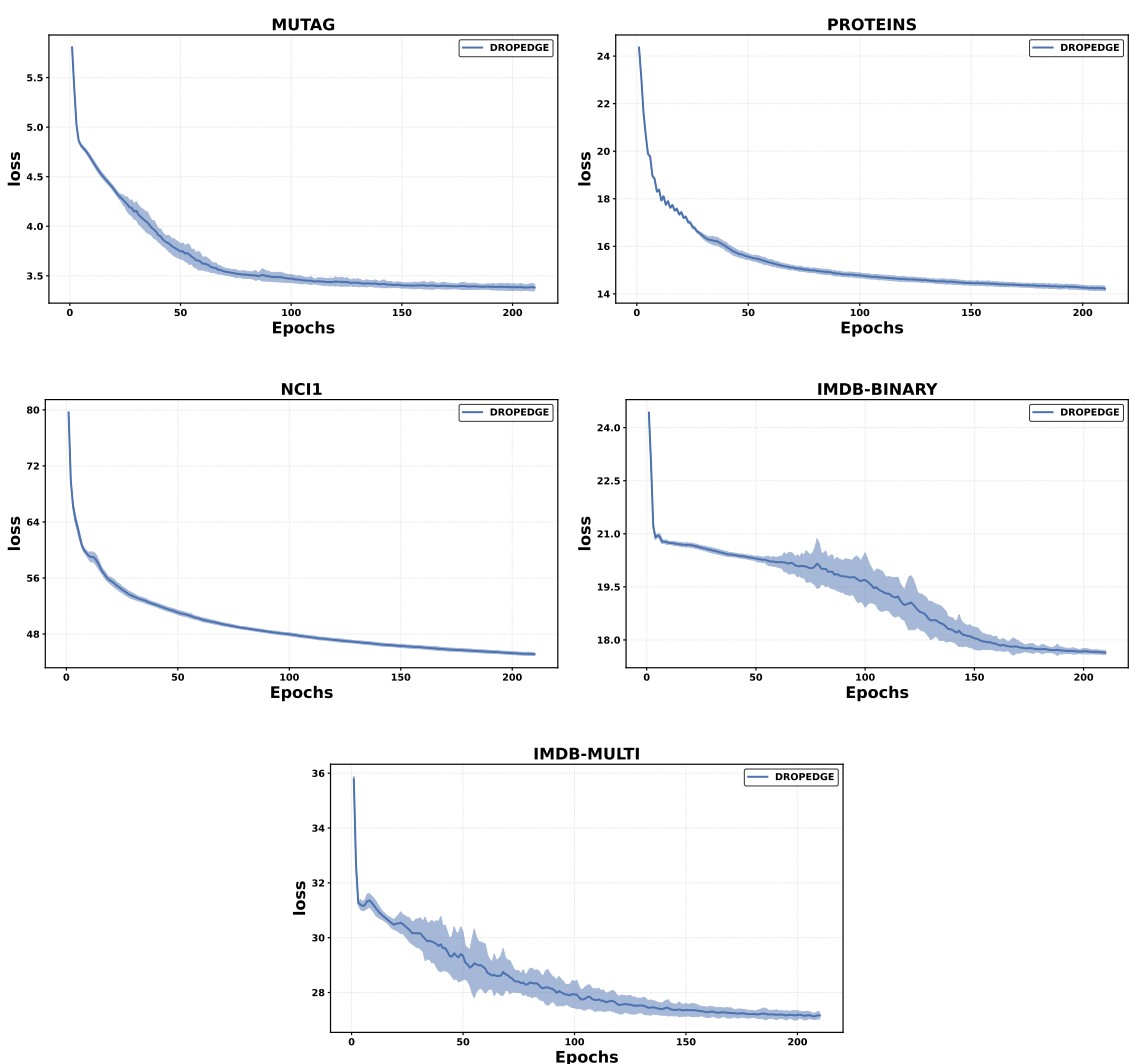

Figure 6: Empirical behavior of InfoNCE loss across different datasets with GRACE and one-layer GCN.

Table 7: Impact of GCN depth on node classification task on the CORA dataset. The best result of each column is in  grey . Metric is accuracy (%).

| MODEL | 4 | 6 | 8 |
|---|---|---|---|
| GBT+DROPEDGE | 83.53± 1.48 | 82.06± 3.45 | 80.88± 1.38 |
| GBT +ADDEDGE | 81.99± 0.79 | 79.04± 1.59 | 79.41± 1.98 |
| GBT+SPAN | 80.39± 2.17 | 81.25± 1.67 | 79.41± 1.87 |
| GRACE+DROPEDGE | 82.35± 1.08 | 82.47± 1.35 | 81.74± 2.42 |
| GRACE +ADDEDGE | 79.17 ±1.35 | 78.80± 0.96 | 81.00± 0.17 |
| GRACE+SPAN | 80.15± 0.30 | 80.15± 0.79 | 75.98± 1.54 |

## E.3 Relationship between spectral cues and performance of EP

Based on the findings obtained from Sec 7.1, it is very likely that spectral information can not be distinguishable enough for good representation learning on the graph. But to more directly answer the question of whether

Table 8: Impact of GCN depth on graph classification task on the MUTAG dataset. The best result of each column is in  grey . Metric is accuracy (%).

| MODEL | 4 | 6 | 8 |
|---|---|---|---|
| GBT+DROPEDGE | $90.74 \pm 2.61$ | $88.88 \pm 4.53$ | $88.88 \pm 7.85$ |
| GBT +ADDEDGE | $94.44 \pm 0.00$ | $94.44 \pm 4.53$ | $94.44 \pm 4.53$ |
| GBT+SPAN | $94.44 \pm 4.53$ | $92.59 \pm 2.61$ | $90.74 \pm 2.61$ |
| GRACE+DROPEDGE | $94.44 \pm 0.00$ | $90.74 \pm 2.61$ | $90.74 \pm 2.61$ |
| GRACE +ADDEDGE | $92.59 \pm 5.23$ | $94.44 \pm 4.53$ | $94.44 \pm 0.00$ |
| GRACE+SPAN | $90.74 \pm 2.61$ | $90.74 \pm 5.23$ | $88.88 \pm 7.85$ |

Table 9: Accuracy of node classification with different GNN encoders on CORA dataset. The best result of each column is in  grey . Metric is accuracy (%).

| MODEL | GCN | GAT |
|---|---|---|
| MVGRL+SPAN | $84.57 \pm 0.22$ | $82.90 \pm 0.86$ |
| MVGRL+DROPEDGE | $84.31 \pm 1.95$ | $83.21 \pm 1.41$ |
| MVGRL +ADDEDGE | $83.21 \pm 1.65$ | $83.33 \pm 0.17$ |
| GBT+SPAN | $82.84 \pm 0.90$ | $83.47 \pm 0.39$ |
| GBT + DROPEDGE | $84.19 \pm 2.07$ | $84.06 \pm 1.05$ |
| GBT + ADDEDGE | $85.78 \pm 0.62$ | $81.49 \pm 0.45$ |
| GRACE + SPAN | $82.84 \pm 0.91$ | $82.74 \pm 0.47$ |
| GRACE + DROPEDGE | $84.19 \pm 2.07$ | $82.84 \pm 2.58$ |
| GRACE + ADDEDGE | $85.78 \pm 0.62$ | $82.84 \pm 1.21$ |
| BGRL + SPAN | $83.33 \pm 0.45$ | $82.59 \pm 0.79$ |
| BGRL + DROPEDGE | $83.21 \pm 3.29$ | $80.88 \pm 1.08$ |
| BGRL + ADDEDGE | $81.49 \pm 1.21$ | $82.23 \pm 2.00$ |

spectral cues and information play an important role in the learning performance of **EP**, we continue to conduct a statistical analysis to evaluate the influence of various factors on the learning performance. The results turn out to be consistent with our claim that spectral cues are insignificant aspects of outstanding performance on accuracy observed in Sec. 6.

### E.3.1   Statistical analyses on key factors on performance of EP

From a statistical angle, we have a few dimensions of factors that can possibly influence learning performance, like the parameters of **EP** (i.e. drop rate $p$ in DROPEDGE or add rate $q$ in ADDEDGE) as well as potential spectral cues lying in the argument graphs. Therefore, to rule out the possibility that spectral cues and information are significant, comparisons are conducted on the impact of the parameters of **EP** in the augmentations versus:

1. The average $L_2$-distance between the spectrum of the original graph (OG) and that of each augmented graph (AUG) which is introduced by **EP** augmentations, denoted as OG-AUG.

2. The average $L_2$-distance between the spectra of a pair of augmented graphs appearing in the same learning epoch when having a two-way contrastive learning framework, like G-BT, denoted as AUG-AUG.

Two statistical analyses have been carried out to argue that the former is a more critical determinant and a more direct cause of the model efficacy. Each analysis was chosen for its ability to effectively dissect and compare the impact of edge perturbation parameters versus spectral changes.

Table 10: Accuracy of graph classification with different GNN encoders on MUTAG dataset. The best result of each column is in grey . Metric is accuracy (%).

| MODEL | GCN | GAT | GPS |
|---|---|---|---|
| MVGRL+SPAN | $93.33 \pm 2.22$ | $96.29 \pm 2.61$ | $94.44 \pm 0.00$ |
| MVGRL+DROPEDGE | $93.33 \pm 2.22$ | $92.22 \pm 3.68$ | $96.26 \pm 5.23$ |
| MVGRL +ADDEDGE | $94.44 \pm 3.51$ | $94.44 \pm 6.57$ | $95.00 \pm 5.24$ |
| GBT+SPAN | $90.00 \pm 6.47$ | $94.44 \pm 4.53$ | $90.74 \pm 5.23$ |
| GBT + DROPEDGE | $92.59 \pm 2.61$ | $94.44 \pm 4.53$ | $94.44 \pm 4.53$ |
| GBT + ADDEDGE | $92.59 \pm 2.61$ | $92.59 \pm 2.61$ | $94.44 \pm 4.53$ |
| GRACE + SPAN | $90.00 \pm 4.15$ | $96.29 \pm 2.61$ | $92.59 \pm 2.61$ |
| GRACE + DROPEDGE | $88.88 \pm 3.51$ | $94.44 \pm 0.00$ | $94.44 \pm 4.53$ |
| GRACE + ADDEDGE | $92.22 \pm 4.22$ | $96.29 \pm 2.61$ | $94.44 \pm 0.00$ |
| BGRL + SPAN | $90.00 \pm 4.15$ | $94.44 \pm 4.53$ | $94.44 \pm 0.00$ |
| BGRL + DROPEDGE | $88.88 \pm 4.96$ | $90.74 \pm 4.54$ | $92.59 \pm 5.23$ |
| BGRL + ADDEDGE | $91.11 \pm 5.66$ | $96.29 \pm 2.61$ | $96.29 \pm 2.61$ |

Due to the high cost of calculating the spectrum of all AUGs in each epoch and the stability of the spectrum of the node-level dataset (as the original graph is fixed in the experiment), we perform this experiment on the contrastive framework and augmentation methods with the best performance in the study, i.e. G-BT with DROPEDGE on node-level classification. Also, we choose the small datasets, CORA for analysis. Note that the smaller the graph, the higher the probability that the spectrum distance has a significant influence on the graph topology.

**Analysis 1: Polynomial Regression.** Polynomial regression was utilized to directly model the relationship between the test accuracy of the model and the average spectral distances introduced by **EP**. This method captures the linear, or non-linear influences that these spectral distances may exert on the learning outcomes, thereby providing insight into how different parameters affect model performance.

Table 11: Polynomial regression of node-level accuracy over drop rate $p$ in DROPEDGE, average spectral distance between OG and AUG (OG-AUG), and average spectral distance between AUG pairs (AUG-AUG). The method is G-BT and the dataset is CORA. The best results are in grey .

| Order of the regression | Regressor | R-squared ↑ | Adj. R-squared ↑ | F-statistic ↑ | P-value ↓ |
|---|---|---|---|---|---|
| 1 (i.e. linear) | Drop rate $p$ | 0.628 | 0.621 | 81.12 | 6.94e-12 |
| | OG-AUG | 0.388 | 0.375 | 30.45 | 1.35e-06 |
| | AUG-AUG | 0.338 | 0.325 | 24.55 | 9.39e-06 |
| 2 (i.e. quadratic) | Drop rate $p$ | 0.844 | 0.837 | 126.9 | 1.14e-19 |
| | OG-AUG | 0.721 | 0.709 | 60.78 | 9.23e-14 |
| | AUG-AUG | 0.597 | 0.580 | 34.88 | 5.16e-10 |

The polynomial regression analysis in Table 11 highlights that the drop rate $p$ is the primary factor influencing model performance, showing strong and significant linear and non-linear relationships with test accuracy. In contrast, both the OG-AUG and AUG-AUG spectral distances have relatively minor impacts on performance, indicating that they are not significant determinants of the model's efficacy.

**Analysis 2: Instrumental Variable Regression.** To study the causal relationship, we perform an Instrumental Variable Regression (IVR) to rigorously evaluate the influence of spectral information and edge perturbation parameters on the performance of **CG-SSL** models. Specifically, we employ a Two-Stage Least Squares (IV2SLS) method to address potential endogeneity issues and obtain unbiased estimates of the causal effects.

In IV2SLS analysis, we define the variables as follows:

- **Y (Dependent Variable):** The outcome we aim to explain or predict, which in this case is the performance of the SSL model.

- **X (Explanatory Variable):** The variable that we believe directly influences Y. It is the primary factor whose effect on Y we want to measure.

- **Z (Instrumental Variable):** A variable that is correlated with X but not with the error term in the Y equation. It helps to isolate the variation in X that is exogenous, providing a means to obtain unbiased estimates of X's effect on Y.

In this specific experiment, we conduct four separate regressions to compare the causal effects of these factors:

1. **(X = AUG-AUG, Z = Parameter):** Examines the relationship where the spectral distance between augmented graphs (AUG-AUG) is the explanatory variable (X) and edge perturbation parameters are the instrument (Z).

2. **(X = Parameter, Z = AUG-AUG):** Examines the relationship where the edge perturbation parameters are the explanatory variable (X) and the spectral distance between augmented graphs (AUG-AUG) is the instrument (Z).

3. **(X = OG-AUG, Z = Parameter):** Examines the relationship where the spectral distance between the original and augmented graphs (OG-AUG) is the explanatory variable (X) and edge perturbation parameters are the instrument (Z).

4. **(X = Parameter, Z = OG-AUG):** Examines the relationship where the edge perturbation parameters are the explanatory variable (X) and the spectral distance between the original and augmented graphs (OG-AUG) is the instrument (Z).

Table 12: IV2SLS regression results for the node-level task. The parameter $p$ refers to the drop rate in DROPEDGE. The experiment comes in pairs for each pair of variables and the better result is marked in grey .

| Variable settings | R-squared ↑ | F-statistic ↑ | Prob (F-statistic) ↓ |
|---|---|---|---|
| ($\mathbf{X}$ = AUG-AUG, $\mathbf{Z} = p$) | 0.341 | 45.77 | 1.68e-08 |
| ($\mathbf{Z} = p$ ,$\mathbf{Z}$ = AUG-AUG) | 0.611 | 47.85 | 9.85e-09 |
| ($\mathbf{X}$ = OG-AUG, $\mathbf{Z} = p$) | 0.250 | 40.22 | 7.51e-08 |
| ($\mathbf{X} = p$, $\mathbf{Z}$ = OG-AUG) | 0.606 | 41.27 | 5.62e-08 |

The IV2SLS regression results for the node-level task in Table 12 indicate that the edge perturbation parameters are more significant determinants of model performance than spectral distances. Specifically, when the spectral distance between augmented graphs (AUG-AUG) is the explanatory variable (X) and drop rate $p$ are the instrument (Z), the model explains 34.1% of the variance in performance (R-squared = 0.341). Conversely, when the roles are reversed (X = $p$, Z = AUG-AUG), the model explains 61.1% of the variance (R-squared = 0.611), indicating a stronger influence of edge perturbation parameter $p$. A similar conclusion can be made when comparing OG-AUG and $p$.

**Summary of Regression Analyses**  The analyses distinctly show that the direct edge perturbation parameters have a consistently stronger and more significant impact on model performance than the two types of spectral distances that serve as a reflection of spectral information. The results support the argument that while spectral information might have contributed to model performance, its significance is extremely limited and the parameters of the **EP** methods themselves are more critical determinants.

### E.4 Comparison with feature augmentation

Although this paper primarily investigates the impact of spectral information and employs simple edge perturbation algorithms for comparison, it is still worthwhile to compare these algorithms with feature augmentation–based graph self-supervised learning methods (Zhao et al., 2022). For node-level tasks, a representative approach is DGI (Velickovic et al., 2019), which uses row-wise shuffling of the node feature matrix X as a feature augmentation strategy. For graph-level tasks, we compare against a more recent work, GCML (Zhang et al., 2024), which introduces message augmentation as a general framework for graph data (feature) augmentation.

Experimental results (Table 13, Table 14) suggest that simple edge perturbation algorithms outperform feature augmentation methods, including the recent GCML. However, we do not claim that edge perturbation is the best possible augmentation method for graph contrastive learning. Rather, our aim is to demonstrate that straightforward topological augmentations can be as effective as—if not better than—many complex, specially designed approaches, thereby offering insights for future research on CG-SSL.

Table 13: Node classification. Results of baselines with '†' are adopted directly from previous works. For **EP**, the result is selected from the best setup for the task. The best results are highlighted in grey . The metric is accuracy (%).

| Model | CORA | CITESEER | PUBMED | PHOTO | COMPUTERS | COAUTHOR-CS | COAUTHOR-PHY |
|---|---|---|---|---|---|---|---|
| EP | 86.51 ± 2.04 | 75.44 ± 0.32 | 87.84 ± 0.37 | 93.68 ± 0.79 | 90.43 ± 0.33 | 93.31 ± 0.05 | 96.06 ± 0.24 |
| DGI† | 82.34 ± 0.64 | 71.85 ± 0.74 | 76.82 ± 0.61 | 91.61 ± 0.22 | 83.95 ± 0.47 | 92.15 ± 0.63 | 94.51 ± 0.52 |

Table 14: Graph classification. Results of baselines with '†' are adopted directly from previous works. For **EP**, the result is selected from the best setup for the task. The best results are highlighted in grey . The metric is accuracy. (%).

| Model | MUTAG | PROTEINS | NCI1 | IMDB-BINARY | IMDB-MULTI |
|---|---|---|---|---|---|
| EP | 94.44 ± 3.51 | 80.64 ± 1.68 | 79.97 ± 2.35 | 76.40 ± 0.48 | 51.73 ± 2.43 |
| GMCL† | 94.09 ± 6.28 | 77.44 ± 3.83 | 82.99 ± 2.06 | 75.60 ± 3.17 | — |

