# OpenReview forum: "Rethinking Spectral Augmentation for Contrast-based Graph Self-Supervised Learning"
_TMLR — Accepted by TMLR_

### Review · Reviewer_5WMb · 2024-12-22

**Summary Of Contributions:**

The paper critiques the importance of spectral augmentation in training contrastive models for learning representations of graphs in a self-supervised way. The authors argue that a simple edge perturbation is more effective than spectral augmentation for generating different views of data while at the same time being more computationally efficient. They perform extensive experiments on multiple node and graph classification datasets to prove their point.

**Audience:**

Yes

**Claims And Evidence:**

Yes

**Requested Changes:**

I would like to see at least one case study (it can be in the appendix) discussing how the edge perturbation augmentation method compares to node feature augmentation techniques.

**Strengths And Weaknesses:**

**Strength**

The paper is well-written and performs multiple experiments on graph and node representation learning datasets. The paper lists the necessary background and details needed to understand the contribution of the paper. It studies whether spectral augmentations[1] are really necessary. The paper identifies the computational cost of using spectral augmentation while performing similarly or lower than adding/dropping edge techniques.

**Weakness**

Even if the paper is well written, the main weakness of the paper is that it doesn't propose any new method. Future research can use it to avoid spectral augmentation, but they could reach this conclusion if they tried a few other augmentation techniques.

The paper shows a theoretical analysis of InfoNCE bounds on Theorem 1. Even though it might be important to show the bounds of the loss, I don't find it convincing enough to show how that is a downside for spectral augmentation. The numerical estimation also gives an estimated number, but it's not stated how that would compare to edge perturbation augmentations. Knowing there is a bound doesn't necessarily mean it's performing worse unless there is a way to compare it with another estimated bound.

The paper doesn't compare the preferred edge perturbation technique with other augmentation techniques. For example, how good is it compared to node feature augmentation methods [2]? Even though the paper focuses on topological augmentation, it would be great to see the result.


**References**

[1] 	Nian Liu, Xiao Wang, Deyu Bo, Chuan Shi, Jian Pei: Revisiting Graph Contrastive Learning from the Perspective of Graph Spectrum. NeurIPS 2022

[2] A Zhao, Tong; Jin, Wei; Liu, Yozen; Wang, Yingheng; Liu, Gang; Graph Data Augmentation for Graph Machine Learning: A Survey, arXiv e-prints, Art. no. arXiv:2202.08871, 2022. doi:10.48550/arXiv.2202.08871.

---

> ### Author Response · Authors · 2025-01-09
> **Thanks for your comments!**
>
> ### 1. Comparison to node feature augmentations.
>
> > Thanks for your comment! We have addressed this in **General Comment 1.1**
> In **Appendix E.4** We have also added a detailed discussion and comparison of edge perturbation against feature augmentation algorithms[1].
>
> ### 2. Connection of Thoerem 1 to spectral augmentations.
>
> > Thank you for your insightful question! We have explained this in **General Comment 1.2**. We also added an empirical study to help validate Theorem 1 to **Appendix D.7**.
>
> ### 3. About the new method.
>
> > Thank you for your comment! Actually, it is a bit out of the scope of our work as this “revisiting” style paper claims to say we don’t need to use spectral augmentation compared to simple topological perturbation. But in another sense, DropEdge and AddEdge can be somehow considered a “new method” people can adopt in the future since they are super simple while having quite good performance in general, compared to other related works, as we include in Tables 2 and 3.
>
> [1] A Zhao, Tong; Jin, Wei; Liu, Yozen; Wang, Yingheng; Liu, Gang; Graph Data Augmentation for Graph Machine Learning: A Survey, arXiv e-prints, Art. no. arXiv:2202.08871, 2022. doi:10.48550/arXiv.2202.08871.

---

### Review · Reviewer_j2VY · 2024-12-29

**Summary Of Contributions:**

This manuscript presents a critical evaluation of the role of spectral augmentation in Contrast-based Graph Self-Supervised Learning. The authors challenge the prevailing belief in the importance of spectral properties for learning performance and propose that simple edge perturbations can achieve comparable or superior results with greater computational efficiency.

**Audience:**

Yes

**Claims And Evidence:**

Yes

**Requested Changes:**

Please see the weaknesses part, and discuss the potential solutions to the questions.

**Strengths And Weaknesses:**

Strengths:
1. The research question in challenging and helpful, demonstrating that pure edge permutations are enough for GCL. This could pave the path for future research directions.
2. The manuscript is well-structured, with clear sections that logically lead the reader through the research process and findings.
3. The theoretical analysis complements the empirical results, offering a more complete understanding of the phenomena observed.
4. The use of multiple baselines and datasets adds robustness to the findings.

Weaknesses and questions:
1. In section 4, authors only use SPAN as spectral augmentation to show the limitations of shallow GNNs while there are many other alternatives. Also, authors only proved that the expressiveness of shallow GNNs using InfoNCE loss are bounded while there are other loss functions as authors mentioned.
2. In figure 1, authors show that the shallow GNNs (1 or 2 layers) usually have better performances than deeper ones. However, in theorem 1, authors claim that sophisticated spectral augmentations may not significantly outperform simple ones in shallow architectures and potential benefits of coupling spectral augmentations with deeper GNN architectures remain an open question. So when increasing the layers, how would the theorem support the experimental findings? If the loss is not bounded strictly like this, why would the performance drop drastically?

---

> ### Author Response · Authors · 2025-01-09
> **Thanks for your comments**
>
> We appreciate your positive feedback and recognition of the significant contribution of our work! For the two weaknesses/requested changes mentioned in your comments, we have addressed them accordingly as follows:
>
> ## 1. Weakness 1
>
> ### 1.1. Only use SPAN to show the limitations of shallow GNNs.
> > Thanks for the comment! Firstly, we claim almost all the existing augmentations need shallow GNN, otherwise, performance will degrade, as results presented in Figure 1, Table 8, and 9. Also. this is consistent with the configuration of all the related works (SPAN[1] uses 2 layers, SpCo[2] has only 1 layer and GASSER[3] uses 1 or 2 layers). What’s more, to comprehensively address the reviewer’s concern, we added the experimental results of SpCo+GRACE for node classifications with respect to the number of GNNs in **Figure 1**, which shows a similar trend that mode performance degrades drastically when the number of layers gets larger.
>
> ### 1.2. Other loss functions
> > Thanks for pointing this out! We admit that Theorem 1 only addresses the boundedness of the InfoNCE objective in shallow GNNs. We chose InfoNCE because it is a widely used objective in contrastive graph self-supervised learning (CG-SSL), but yes, we do not claim this result automatically extends to all other possible loss functions. Investigating alternative loss functions is an interesting direction, yet beyond our current scope. We aimed to provide initial theoretical insights rather than a universal proof covering every contrastive objective.
>
> We have included this part in **Appendix D.6.3**.
>
> ## 2. Weakness 2 (Deep GNN encoder)
>
> > Thank you for the detailed comment! In response to this question, we note that Theorem 1 is proposed specifically for shallow GNNs and does not claim to directly extend to deeper architectures. Nonetheless, our empirical results indicate that deeper GNNs can exhibit degraded performance in practice, which is consistent with real-world setups for spectral augmentation approaches as well (as we mentioned in detail number in the comment for the comment for Weakness 1 above). This outcome is not surprising, since shallow GNN encoders are widely adopted by various methods in the domain of CG-SSL, mostly for the sake of empirical performance. At the same time, we acknowledge that more work, especially from the theoretical aspect, can be done to explore how deeper GNN architectures might be effectively coupled with spectral augmentations, as the interactions in that setting are far more complex than those considered in our current theoretical result for shallow layers.
>
> We have included this part in **Appendix D.6.3**, too.
>
> [1] Lin et al. Spectral Augmentation for Self-Supervised Learning on Graphs. ICLR 2023.
> [2] Liu et al. Revisiting Graph Contrastive Learning from the Perspective of Graph Spectrum. NeurIPS 2022.
> [3] Yang et al. Augment with Care: Enhancing Graph Contrastive Learning with Selective Spectrum Perturbation. Preprint

---

### Review · Reviewer_ZUcv · 2025-01-04

**Summary Of Contributions:**

The paper examines the necessity and effectiveness of spectral augmentation in the context of contrastive graph self-supervised learning (CG-SSL). It is shown that shallow GNNs consistently outperform deeper ones in CG-SSL, and theoretical bounds on the InfoNCE loss are provided. It is further demonstrated empirically that simple edge perturbations match and sometimes surpass the performance of spectral augmentation methods while being computationally more efficient. Ablation experiments and a statistical analysis support the conclusion that spectral information is not a key factor in the success of edge perturbation methods, advocating for simpler, more efficient augmentation strategies in CG-SSL.

**Audience:**

Yes

**Claims And Evidence:**

Yes

**Requested Changes:**

* Clarify the presentation of Theorem 1 and its relationship to spectral augmentations
* Potentially add an experimental comparison of the InfoNCE bound from Theorem 1 and the actual loss values during training
* Fix grammar/spelling issues

**Strengths And Weaknesses:**

**Strengths**

Overall, this paper presents significant observations about the (in-)effectiveness of spectral augmentations. The experiments consider four different CG-SSL frameworks as well as node- and graph-level tasks and show that the main observations are consistent across these. The additional ablation studies in Section 7 are also insightful to the reader.

**Weaknesses**

I think the presentation of Theorem 1 and its connection to spectral augmentations could be improved. It would be useful to provide an intuition for how $\epsilon$ and $\epsilon'$ depend (asymptotically) on the model and augmentation hyperparameters ($k,\delta,\dots$). While information about this is provided in formal detail in the Appendix, I think the presentation in the main section could at least provide a general idea of how the upper and lower bounds are actually related to these hyperparameters. The motivation for this theorem and how it is connected to spectral augmentations is also somewhat vague. It is not fully clear to me why the fact that the InfoNCE loss is constrained to a particular band of values has significant implications for whether or not spectral augmentations are useful. As far as I understand, graph augmentation methods (spectral or otherwise) are generally not motivated by the specific value that a contrastive loss reaches during training but by the graph perturbations that the embeddings are meant to be resilient to. While the empirical study shows that spectral perturbations seem to not be significantly better than simpler choices, I currently find it unclear how this theorem relates to those observations. Finally, it would be interesting to compare the range of the InfoNCE loss predicted by this theorem to the loss actually observed during training on a real dataset with given hyperparameters.

Throughout the paper, there are also a few grammatical mistakes / strange phrasing choices that could be improved, such as:
* "Can GNN encoders learn spectral information from augmented graphs produced edge perturbations" (Page 9, **by** edge perturbations?)
* "Are spectrum in spectral augmentation necessary?" (Page 9)
* ...a bunch of benchmarks..." (Page 5, informal)

---

> ### Author Response · Authors · 2025-01-09
> **Thanks for your comments!**
>
> We appreciate your positive feedback and recognition of the insightfulness of our work! For the three weaknesses/requested changes mentioned in your comments, we have addressed them accordingly as follows:
>
> ### 1. Clarify the presentation of Theorem 1 and its relationship to spectral augmentations
>
> > Thanks for the comment! We have addressed this in **General Comment 1.2**.
>
>
> ### 2. Add an experimental comparison of the InfoNCE bound from Theorem 1 and the actual loss values during training.
>
> > We sincerely thank the reviewer for this insightful suggestion. We have added empirical validation of our theoretical InfoNCE bounds in **Appendix D.7**. Our experiments across multiple datasets using GRACE with one- or two-layer GCN encoder (edge dropping rate = 0.3, 100 independent runs) demonstrate that:
> >> 1). The InfoNCE loss exhibits remarkable stability on MUTAG and PROTEINS datasets, with minimal variations despite diverse spectral changes from random edge perturbations.
>
> >> 2). While we observe larger fluctuations in the InfoNCE loss during the middle stages of training on the dense IMDB dataset(variations >1.0), these fluctuations consistently converge to a narrow band during the final stages.
>
> > These results strengthen our theoretical findings by showing that even with diverse spectral variations from edge perturbations, the InfoNCE loss remains well-bounded.
>
> ### 3. Grammar issues.
>
> > Thanks for pointing that out. We have revised everything in the updated manuscript.

---

### Review · Reviewer_ykmD · 2025-01-04

**Summary Of Contributions:**

This paper questions the importance of spectral augmentation in CG-SSL and shows that simple edge perturbations can match or outperform it while being more efficient. It also provides theoretical InfoNCE bounds and supports its claims with extensive experiments and ablation studies.

**Audience:**

Yes

**Claims And Evidence:**

Yes

**Requested Changes:**

Please refer to weaknesses

**Strengths And Weaknesses:**

Strengths:
1. It challenges widely held assumptions about spectral augmentation.
2. It offers solid experimental validation across multiple datasets and tasks.
3. It combines theoretical insights with practical findings.


Weaknesses:

1. How about comparing edge perturbations with other types of augmentations, like node feature perturbations.
2. Can author provide more detailed connections between theorem 1 and the spectral augmentations?

---

> ### Author Response · Authors · 2025-01-09
> **Thanks for your comments**
>
> We appreciate your positive feedback on the work! For the two weaknesses (also requested changes) mentioned in your comments, we have addressed them accordingly as follows:
>
> ### 1. Node feature augmentations/perturbations.
> > Thank you for your comment! We have addressed this in **General Comment 1.1** and added relevant revision in **Appendix E.4**. We have also added a detailed discussion and comparison of edge perturbation against feature augmentation algorithms[1].
>
> ### 2. More detailed connections between theorem 1 and the spectral augmentations.
> > Thanks for the comment! We have addressed this in **General Comment 1.2**. We also added an empirical study to help validate Theorem 1 to **Appendix D.7**.
>
>
> [1] A Zhao, Tong; Jin, Wei; Liu, Yozen; Wang, Yingheng; Liu, Gang; Graph Data Augmentation for Graph Machine Learning: A Survey, arXiv e-prints, Art. no. arXiv:2202.08871, 2022. doi:10.48550/arXiv.2202.08871.

---

### Author Response · Authors · 2025-01-09
**General Comments from Authors**

## 1. Overall comments
We sincerely thank all the reviewers and the AE for the effort they have put into reviewing our work.  We have tried our best to address all concerns and requested changes raised by the reviewers and add revisions accordingly to the paper for each of them (we have highlighted the pointer to them in **bold** in rebuttal here and **blue** color in the revised version of the paper). From the comments, we notice there are two questions shared in common so we address them here as General Comments 1.1 and 1.2. For convenience, we give a code name to each reviewer as follows:

Reviewer ykmD: [A]

Reviewer ZUcv: [B]

Reviewer j2VY: [C]

Reviewer 5WMb: [D]

## 1.1 Comparison to node feature augmentations (from Reviewer [A] and [D] ):

> We appreciate Reviewer [A] and Reviewer [D]’s suggestion to add a comparison against node feature augmentation CG-SSL algorithms to our paper. To address this, we added a section in **Appendix E.4**, where we selected two representative feature augmentation CS-SSL algorithms, DGI for node-level tasks, and GMCL for graph-level tasks. Experimental results indicate that simple edge perturbations also outperform these feature augmentation algorithms, especially since GMCL is a very recent work. We appreciate the idea and the addition of this section to our revision is making our claim more persuasive.


## 1.2 Connection of Theorem 1 to Spectral Augmentation (Reviewer [A], [B], [D]):

> As Reviewers [A], [B], and [D] have some questions related to the positioning of Theorem 1, we have the following global comments to address those questions. We have also added a detailed discussion in **Appendix D.6**.

### Positioning of Theorem 1 (**Appendix D.6.1**)
> We appreciate the reviewers’ request for clarity regarding Theorem 1 and its relationship to spectral augmentations. This connection is indirect due to the inherent difficulty of providing direct theoretical proof that simultaneously addresses spectral information, the InfoNCE training objective, and downstream tasks in CG-SSL. Nevertheless, Theorem 1 offers a useful lens by showing that under common assumptions, the InfoNCE objective remains bounded within a finite range regarding the magnitude of local topological perturbation. Per request of Reviewer [B] (i.e. ZUcv), we also carry out empirical validation of our theoretical InfoNCE bounds in **Appendix D.7**.

### Connection to spectral augmentations (**Appendix D.6.2**)
> Because spectral augmentations primarily target the graph’s eigenstructure, they may not necessarily push the loss beyond these inherent bounds, where the loss is critical to the optimization of the CG-SSL and eventually to the performance of downstream tasks. Therefore, we view Theorem 1 as an initial theoretical insight rather than a definitive statement on the value of spectral augmentations. By highlighting that the InfoNCE loss is constrained, it adds context to why complicated augmentation schemes and simpler approaches may converge to similar performance levels. We remain optimistic that further theoretical and empirical investigations can deepen the understanding of how spectral perturbations, and graph augmentation, interact with contrastive objectives in CG-SSL.

---

### Decision · Action_Editor_vzw3 · 2025-02-12

**Recommendation:** Accept with minor revision

**Comment:**

The paper presents a well-executed empirical and theoretical analysis that challenges the necessity of spectral augmentations in contrast-based graph self-supervised learning, demonstrating that simple edge perturbations achieve comparable or superior performance with lower computational costs. While the connection between theoretical and empirical findings could be further refined, the authors have sufficiently addressed reviewer concerns through additional experiments and clarifications, making this a strong contribution to the field. I recommend acceptance with minor revisions, contingent on the authors incorporating the revisions outlined in their rebuttal.

**Audience:**

Yes

**Claims And Evidence:**

Yes

---

> ### Author Response · Authors · 2025-02-16
> **Camera-ready revision**
>
> Dear AE,
>
> Thank you for your thoughtful feedback and for recommending our paper for acceptance. We appreciate the constructive review process and the opportunity to strengthen our work. As requested, we have incorporated all the revisions outlined in our rebuttal (in blue in the previous revision) and uploaded the camera-ready version.
>
> Best regards,
> Authors of Submission 3815